# Reprogramming of *cis*-regulatory networks during skeletal muscle atrophy in male mice

Hongchun Lin[1,2], Hui Peng [1,2] ✉, Yuxiang Sun [1], Meijun Si[3], Jiao Wu[2], Yanlin Wang[4], Sandhya S. Thomas[2], Zheng Sun [5] & Zhaoyong Hu [2] ✉

A comprehensive atlas of cis-regulatory elements and their dynamic activity is necessary to understand the transcriptional basis of cellular structure maintenance, metabolism, and responses to the environment. Here we show, using matched single-nucleus chromatin accessibility and RNA-sequencing from juvenile male *C57BL6* mice, an atlas of accessible chromatin regions in both normal and denervated skeletal muscles. We identified cell-type-specific cis-regulatory networks, highlighting the dynamic regulatory circuits mediating transitions between myonuclear types. Through comparison of normal and perturbed muscle, we delineated the reprogramming of *cis*-regulatory networks in response to denervation, described the interplay of promoters/enhancers and target genes. We further unveil a hierarchical structure of transcription factors that delineate a regulatory network in atrophic muscle, identifying ELK4 as a key atrophy-related transcription factor that instigates muscle atrophy through TGF-β1 regulation. This study furnishes a rich genomic resource, essential for decoding the regulatory dynamics of skeletal muscle in both physiological and pathological states.

Skeletal muscle, which accounts for about 40% of body mass, plays a crucial role in glucose/lipid metabolism, protein storage, movement, and posture. Its metabolic status has a significant impact on health and the prognosis of chronic diseases, as muscle wasting increases the risk of morbidity and mortality in conditions such as ageing, chronic kidney disease, and neurogenic myopathy[1,2]. Skeletal muscle is composed of myofibers and resident cells, including FAPs, satellite cells, vascular cells, and macrophages. Myofibers are multiple-nuclei cells that can be classified based on their expression of myosin heavy chain (Myh) isoforms and energy metabolism properties[3]. In response to environmental changes, myofibers undergo fiber-type switching and remodeling, which is driven by changes in their transcriptional profile[4]. Similarly, muscle-resident cells experience dramatic transcriptomic changes and participate in extensive intercellular communication that contributes to the phenotypic changes in muscle fibers[5]. While whole-muscle assays have provided valuable insights into gene regulation

involved in muscle development and myofiber remodeling[6], they have limited ability to characterize networks in a cell-type-specific manner.

The recent development of single-cell or single-nucleus assays such as snATAC-seq and snRNA-seq provides additional approaches for understanding the mechanisms underlying muscle cell identities[7,8]. These assays can define the gene regulatory logic that underlies cell identity, cell type-specific gene control, differentiation, and pathogenesis[9]. snATAC-seq identifies the regulatory elements and transcription factor binding sites that maintain myofiber type specificity, while snRNA-seq improves our understanding of muscle development and degeneration in mice and humans[10,11]. Gene regulatory networks involve the interplay between transcription regulators and target genes, with *cis*-regulation being the predominant mode of gene regulation in mammalian cells[12]. This process involves specific DNA sequences, called *cis*-regulatory elements (CREs), and the transcription factors (TFs) that bind to them, triggering transcription of a

[1]Nephrology Division, the Third Affiliated Hospital of Sun Yat-sen University, Guangzhou 510630, China. [2]Nephrology Division, Department of Medicine, Baylor College of Medicine, Houston, TX 77030, USA. [3]Department of Nephrology, Guangdong Provincial People's Hospital, Guangzhou 510080, China. [4]Division of Nephrology, Department of Medicine, University of Connecticut School of Medicine, Farmington, CT, USA. [5]Endocrinology Division, Department of Medicine, Baylor College of Medicine, Houston, TX 77030, USA. ✉e-mail: pengh@mail.sysu.edu.cn; zhaoyonh@bcm.edu

neighboring gene by assembling transcriptional machinery on the transcription start site[13,14]. These interactions result in distinct gene expression programs that confer cell type specificity and specific responses to environmental changes[12]. While detailed maps of CREs and TFs are necessary for understanding cell function, pathological processes, and disease treatment, a comprehensive atlas that emphasizes the properties and reprogramming of *cis*-regulation under pathologic conditions has yet to be established for skeletal muscle.

In this study, we present a single-cell *cis*-regulatory atlas of normal gastrocnemius muscles and those with neurogenic atrophy in 12-week-old male *C57BL6* mice. By integrating snATAC-seq and snRNA-seq assays, we examined chromatin accessibility and nuclear gene expression across 11 cell types, encompassing ~76,000 nuclei. We identified the *cis*-regulatory logic underlying cell identity and myonuclear transition between fast and slow myofibers. In addition, we investigated the chromatin accessibility profiles of denervated muscle, delineated the reprogramming of the *cis*-regulatory network in response to a catabolic stimulus, and constructed a transcription factor hierarchy that defines an atrophy-related responsive circuitry in muscle. Finally, we investigated the role of ELK4, a highly hierarchically ranked transcription factor, in the pathogenesis of muscle atrophy induced by denervation.

## Results

### An atlas of *cis*-regulatory networks in normal skeletal muscle

To elucidate the gene regulatory network in skeletal muscle at a single-nucleus resolution, we performed snRNA-seq and snATAC-seq on gastrocnemius (GAS) tissue from normal and denervated (14 days) male mice. In parallel, we performed bulk RNA-seq and H3K27ac ChIP-seq in mouse GAS muscles under matched conditions (Fig. 1a). We constructed an atlas of snRNA-seq for both normal and denervated muscles (Fig. 1b) and integrated it with the snATAC-seq atlas to annotate each cluster, resulting in cell-type-specific, accessible chromatin regions (ACRs) profiles for normal and denervated skeletal muscle (Fig. 1c, Supplementary Fig. 1).

Since a comprehensive *cis*-regulatory landscape has not yet been established in normal mouse skeletal muscle, we identified cell-type-specific CREs by utilizing the Signac R package to analyze differential ACRs (or differential peaks) among eight cell types in normal muscles. In total, we obtained 6297 differential ACRs (min.pct = 0.2, lfc > 0.25, $p < 0.05$), the majority of which displayed cluster-restricted patterns, as demonstrated by various loci of signature genes (Fig. 1d, Supplementary data 1). As chromatin accessibility at TSSs is a well-known prerequisite for transcription, we focused our analysis on the differential ACRs near TSSs (Supplementary Fig. 2a). We observed that a majority of ACRs (>60%) located in proximal promoter regions (within 3 kb of TSS) and distal enhancer regions (>3 kb from TSS), followed by intron and exon regions (Supplementary Fig. 2b). These results indicate that the activation of promoters and enhancers occurs in a cell-type-specific manner. The cell-type-specific gene accessibility we observed was consistent with cell-type-specific gene expression, as demonstrated by the co-localization of accessibility with expression of *Trim63* and *Prkag3* in myofibers, *Fmod* in MTJ, *Chodl* in *MuSCs*, *Scara5* in FAPs, *Nrros* in macrophages, and Tie1 in endothelial cells (Supplementary Fig. 2c, d). We further validated the expression and anatomical position of selected genes in normal GAS muscles through immunostaining (Fig. 1e). Gene ontology (GO) enrichment analysis further revealed that the genes associated with these differential ACRs are relevant to the functions of each cell type, with enrichments of terms such as "Structural constituent of muscle" and "Voltage-gated channel activity" in myonuclei, "Laminin binding" in MTJ, "Notch binding" in MuSCs, and "Cytokine receptor activity" in FAPs (Supplementary Fig. 2e). These results signify that chromatin accessibility determines the transcription and biological function of different cell types in muscle.

To identify the key transcription factors (TFs) that interact with cell-type-specific CREs, we performed motif enrichment analysis on the differential ACRs using ChromVar[15]. This analysis revealed 579 motifs with best-matched TFs, which likely determine cell type specificity in muscle (Supplementary data 2). Figure 1f shows several cell-type-specific enriched motifs and their corresponding TFs, such as MA0796.1/TGIF1, MA1641.1/MYF5, and MA0841.1/NFE2, which are known modulators in muscle cells[16–18]. Notably, using Tn5 enrichment analysis, a method designed to pinpoint ACRs targeted by TF binding, we observed a "summit depletion in peaks" across all selected TF motifs. This indicates that the physical binding of TF to the motif impedes the integration of Tn5 transposase (Fig. 1g and Supplementary Fig. 2g).

Co-accessibility between ACRs can recapitulate the chromatin interactions and *cis*-regulatory circuits in vivo[19]. Accordingly, we employed *cis*-co-accessibility network (CCAN) analysis to predict *cis*-regulatory chromatin interactions in normal muscles[19,20]. We identified 53,212 *cis*-regulatory links covering promoter, distal, and intragenic regions in normal muscle (co-accessibility score >0.2) (Fig. 1h). We observed that many distal region peaks exhibit strong co-accessibility with promoter regions near TSSs, suggesting that they are likely candidate enhancers for the promoter of vicinal genes. Since enhancers are usually marked by H3K27ac, we surveyed our H3K27ac ChIP-seq datasets and found that most of these distal ACRs feature either immediate or flanking H3K27ac enrichment. For example, our analysis was able to infer several candidate enhancers, such as for *Myh7*, *Chodl*, and *Car3* (Fig. 1i). Besides the discovery of promoter and distal enhancers, we also identified many intragenic ACRs that have strong links with promoter or enhancer regions (Fig. 1h). In summary, our integrated *cis*-regulatory analysis with the snATAC-seq and scRNA-seq datasets provided insights into cell-type-specific promoters, candidate enhancers, TFs, and their possible interactions, revealing the *cis*-regulatory circuits present in normal skeletal muscle.

### *cis*-Regulatory dynamics governing myonuclear transition

Evidence suggests the existence of switching between slow (type IIa/IIx) and fast (type IIb) myofiber types in adult skeletal muscle, possibly triggered by factors such as nerve activity, exercise, or hormonal influences[4,21]. We sought to map the *cis*-regulatory network that maintains this dynamic balance under normal condition. To achieve this, we organized myonuclei along a pseudotime axis based on gene expression profile similarities, unveiling a continuous trajectory from type IIa to type IIb myonuclei (Fig. 2a). We subsequently examined the fraction of myonuclei expressing known fiber-type marker genes (*Myh2*, *Myh1*, and *Myh4*), and sequenced them in accordance with transition pseudotime. This analysis led to the observation of a pseudotime-dependent decrease in slow-myosin (*Myh2*)-positive nuclei and a shift towards *Myh1*-positive nuclei, followed by an increase in fast-myosin (*Myh4*)-positive nuclei (Fig. 2b). Notably, this transition produced myonuclei expressing both slow and fast Myhs, aligning with the presence of hybrid myofibers in normal muscle[22].

In addition, we identified changes in the expression of energy homeostasis genes, such as *Ckm* and *Mb*, in relation to pseudotime. This suggests that energy metabolism adapts during this dynamic myonuclear transition (Fig. 2b, Supplementary Fig. 3a). We further investigated the role of nuclear-coding mitochondrial genes in this transition by annotating myonuclei according to the average expression of 1013 mitochondrial genes along the trajectory (Fig. 2c, Supplementary Fig. 3b). Reflecting their high reliance on oxidative metabolism, the majority of type IIa and/or type IIx nuclei showed a higher expression of mitochondrial genes, which progressively decreased in type IIb nuclei as pseudotime advanced, as exemplified by the decline expression of *Pgc1a*, *Mfn1*, *Opa1*, and other genes (Fig. 2d, Supplementary Fig. 3c). Thus, transcriptomic trajectory analysis in myonuclei suggests that the dynamic equilibrium of myofiber-

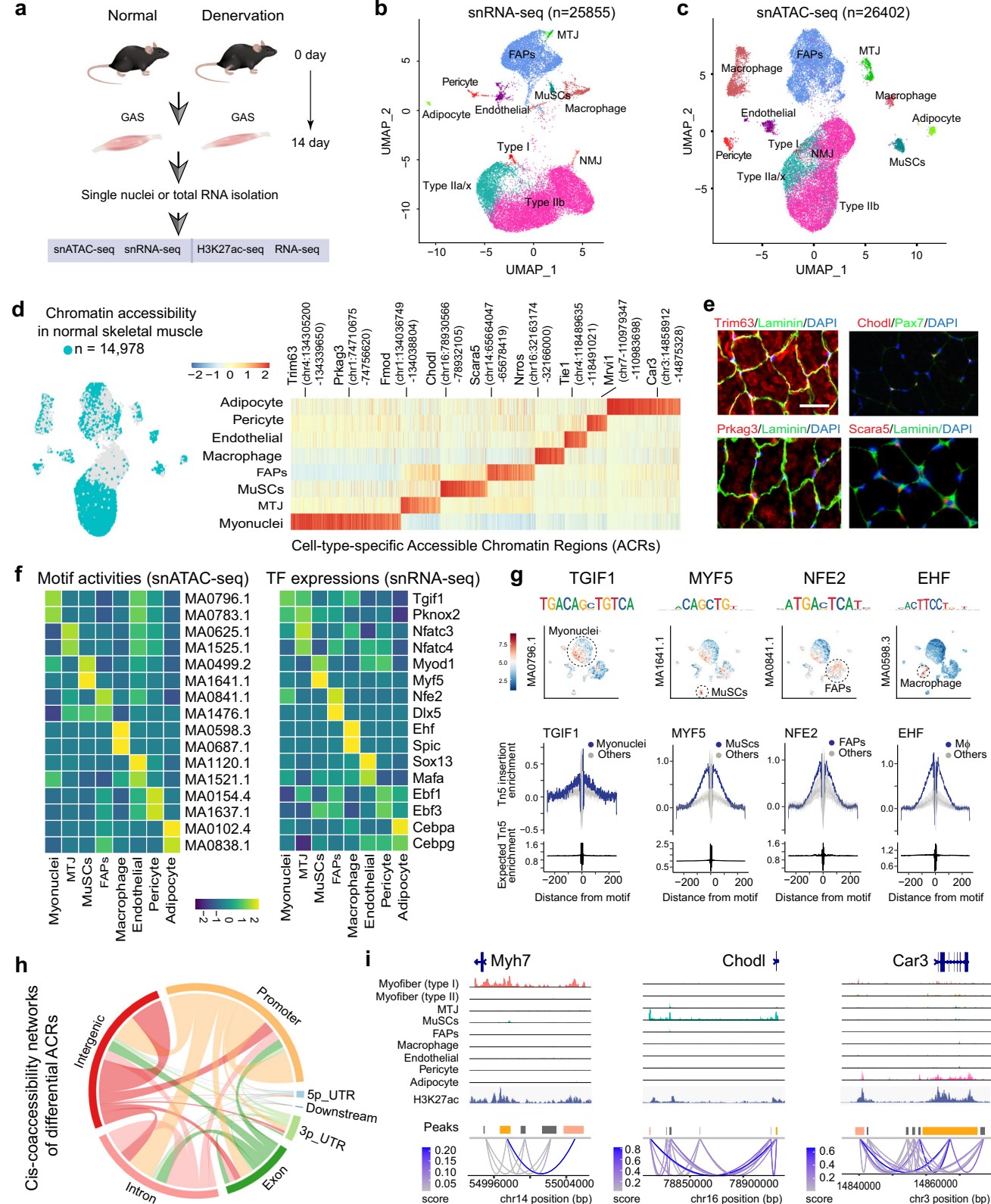

type configuration in normal muscle may be attributable to continuous transcriptional changes in myonuclei.

To corroborate the observed transcriptomic trajectory with changes in DNA accessibility, we conducted a myonuclear trajectory analysis using our snATAC data. We identified significant alterations in accessible ACRs, TF motifs, and TF loci across the pseudotime trajectory (Fig. 2e). We noted 13,134 pseudotime-associated differential

ACRs in type II myonuclei and a global decrease in chromatin accessibility with pseudotime (Fig. 2f). The accessibility status of genes representing different pseudotime periods, including *Ldlrad3*, *Rps6kc1*, and *Dapk1*, was confirmed by their mRNA expression throughout the myonuclear transition (Fig. 2g). In addition, we determined the anatomical expression of their corresponding proteins in GAS muscles using immunostaining. Specifically, Ldlrad3 was

**Fig. 1 | An atlas of *cis*-regulatory networks in normal skeletal muscle.**
**a** Experimental scheme. **b** Uniform Manifold Approximation and Projection (UMAP) plot presents 11 nuclear clusters derived from snRNA-seq data of GAS muscle nuclei in three normal and three denervated mice ($n = 25,855$ nuclei). Clusters are distinguished by color and labeled based on their nuclear identities. **c** UMAP visualization of snATAC-seq data from GAS muscle nuclei of three normal and three denervated mice ($n = 26,402$ nuclei), annotated based on chromatin accessibility to highlight accessible chromatin regions (ACRs). **d** Left: UMAP plot displaying ACRs in normal muscles. Right: Heatmap illustrates the average count of Tn5 insertion sites within DARs for each nuclear type in normal GAS muscle. Signature genes are highlighted at the top, with additional signature gene loci provided in Supplementary Data 1. **e** Representative immunostaining images of proteins encoded by selected genes (*Trim 63*, *Chodl*, *Prkag3*, and *ScaraS*) in the GAS muscle of a normal mouse. Scale bar: 25 μm. **f** Z-score heatmaps display motif activity (left) and the associated TF gene expression (right) for each nuclear type. For the complete list of all 579 motifs and their best-matched TFs, please see Supplementary Data 2. **g** UMAP plots display the activity of cell-type-specific motifs for four selected transcription factors (TFs) in the upper panel, accompanied by their respective footprint plots in the lower panel. The footprint lines are color-coded based on cell identity. **h** *Cis*-coaccessibility networks (CCANs) visualizing interactions among cell-type-specific ACRs. Source data are provided as a Source data file. **i** Aggregated cell-type chromatin accessibility, H3K27ac signal, and CCANs around selected gene loci. Orange indicates the promoter region, while pink highlights distal intergenic regions. Links represent co-accessibility between ACRs, with color density indicating the strength of interaction.

predominantly expressed in Myh2 positive (type IIa) myofibers; Rps6kc1 was typically located in Myh2, Myh7, and Myh4 negative (type IIx) myofibers; and Dapk1 was primarily found in Myh4 positive (type IIb) myofibers (Fig. 2h). Despite these observations, their precise roles in skeletal muscle warrant further investigation. To delineate the *cis*-regulatory elements governing the dynamic myonuclear transition, we identified changes in the accessibility of TF genes and TF binding motifs along the transition path. We analyzed motif enrichment dynamics in snATAC-seq data using ChromVAR and computed the motif activity score for each myonucleus. This revealed 361 TF motifs and 220 TF loci in myonuclei and ordered their normalized activity scores or expression by pseudotime. The results exhibited a strong concordance of time-dependent changes in the accessibility of TFs and their corresponding motifs across the transition path (Fig. 2i), suggesting the chromatin accessibility status of TF motifs and cognate TF genes could elucidate the regulatory rules governing myofiber-type dynamics. In summary, our findings indicate that myonuclear-type dynamics is a trajectory-dependent process. This process involves a dynamic interplay of chromatin organization and TF binding, thus forming complex combinatorial circuits that underpin the regulation of myofiber types.

## Characterization of the cell-type-specific regulatory landscape in denervated muscle

To investigate how chromatin organization and *cis*-regulation of skeletal muscle adapt to a catabolic environment, we analyzed the chromatin accessibility in denervated muscle, a model of high clinical relevance that reflects the adaptive responses of protein metabolism, energy metabolism, and tissue remodeling during muscle atrophy[23]. After 14 days of sciatic denervation, the mice exhibited significant myofiber atrophy in GAS muscles accompanied by changes of myonuclear shape (Fig. 3a, Supplementary Fig. 4a). Transmission electron microscopy (TEM) revealed that denervation causes prominent nucleoli and decreased heterochromatin in myonuclei (Fig. 3a), suggesting that denervation causes profound remodeling of the chromatin architecture in myonuclei[24]. Indeed, the distribution of cell-type-specific ACRs as visualized by UMAP shows clear differences between normal and denervated muscles, especially in myonuclei (Fig. 3b). Compared with normal muscles, we found denervated GAS muscles to have an overall increase in ACRs, discretely distributed over all 21 chromosomes (Fig. 3c, d, Supplementary Fig. 4b). Using ChIPseeker[25], we analyzed ACR localization in a cell-type-specific manner, which revealed an obvious decrease in the percentage of promoter region peaks (<3 kb from TSS) for denervated myofibers compared to normal myofibers (15.12% vs. 38.75%). In contrast, distal and intragenic ACRs were increased in denervated myonuclei (36.18% vs. 21.68% and 48.55% vs. 39.56%, respectively), indicating that *cis*-regulation mediated by distal or intragenic ACRs is intensified during denervation (Fig. 3e). In addition, many other types of muscle-resident cells show less changes between normal vs. denervated (MuSCs and Adipocytes), or changes in

the opposite direction (FAPs and Macrophages) when compared to the differences shown by myonuclei (Fig. 3e).

To investigate the cell-type-specific *cis*-regulatory programs involved in adapting to denervation, we analyzed differentially accessible regions (DARs) between normal and denervated muscles for each annotated cell type ($p < 0.05$, log2FC > 0.25) (Fig. 3f). Overall, DARs were more frequently detected in myonuclei than in other cell types, suggesting this population undergoes tremendous reprogramming of *cis*-regulation upon denervation. Since proximal CREs (promoters) interact with distal CREs (enhancers) to form a loop architecture mediated by CCCTC-binding factor (CTCF) that regulates downstream gene expression[26], we using co-accessibility analysis to link distal DARs to nearby proximal DARs and identify their target genes in each cell type (median distance, 88.7 kbp, Supplementary data 3). In myonuclei, we observed that distal DARs were associated with the expression of myofiber-signature genes or muscle atrophy-related genes (Fig. 3g). For example, several distal DARs with increased accessibility in normal myofibers were linked to the promoter region of *Myh4* (fast type myosin heavy chain), for which accessibility and gene expression were suppressed in denervated GAS muscles (Fig. 3h, left panel). Conversely, distal DARs with increased accessibility in denervated myofiber were linked to the promoter region of *Sh3d19*, a regulator of metalloproteases and cytoskeletal organization[27]. Although *Sh3d19* expression was robustly increased in denervated muscle, its role in regulating muscle atrophy is unexplored (Fig. 3h, right panel). We further performed functional annotation of myonuclear DAR-linked genes using GO biological process terms. This revealed the decreased DARs in denervated muscle to be mainly enriched in the regulation of voltage-gate ion activity, carbohydrate metabolism, and skeletal muscle structural components. On the other hand, those DARs increased upon denervation were predominantly enriched in terms relating to cytoskeletal and actin filament organization, extracellular matrix organization, ubiquitin conjugating enzyme binding (Fig. 3i). These results suggest that DARs in denervated muscles are related to loss of muscle-specific function and gain of dedifferentiation capacity.

Considering FAPs specifically, a total of 760 DARs were identified, ~50% in distal regions and <20% in promoter regions (Fig. 3e, f). Co-accessibility analysis of these DARs revealed them to be associated with genes involved in distinct biological processes (Fig. 3j). Namely, DARs with higher accessibility in denervated FAPs were linked to genes involved in early morphogenesis and lipoprotein metabolism. For example, the expression of *Apod*, which functions as an extracellular secreted protein to modulate steroid hormone transportation and has recently been suggested as a marker of muscle atrophy[28], was strongly induced by denervation and this response was clearly regulated by a distal region detected by co-accessibility analysis (Fig. 3k, left). Conversely, DARs with reduced expression of their linked genes in denervated FAPs were adjacent to genes involved in biological processes such as phospholipid metabolism. For example, we detected lower co-accessibility associated with decreased expression of *ATP8A1*,

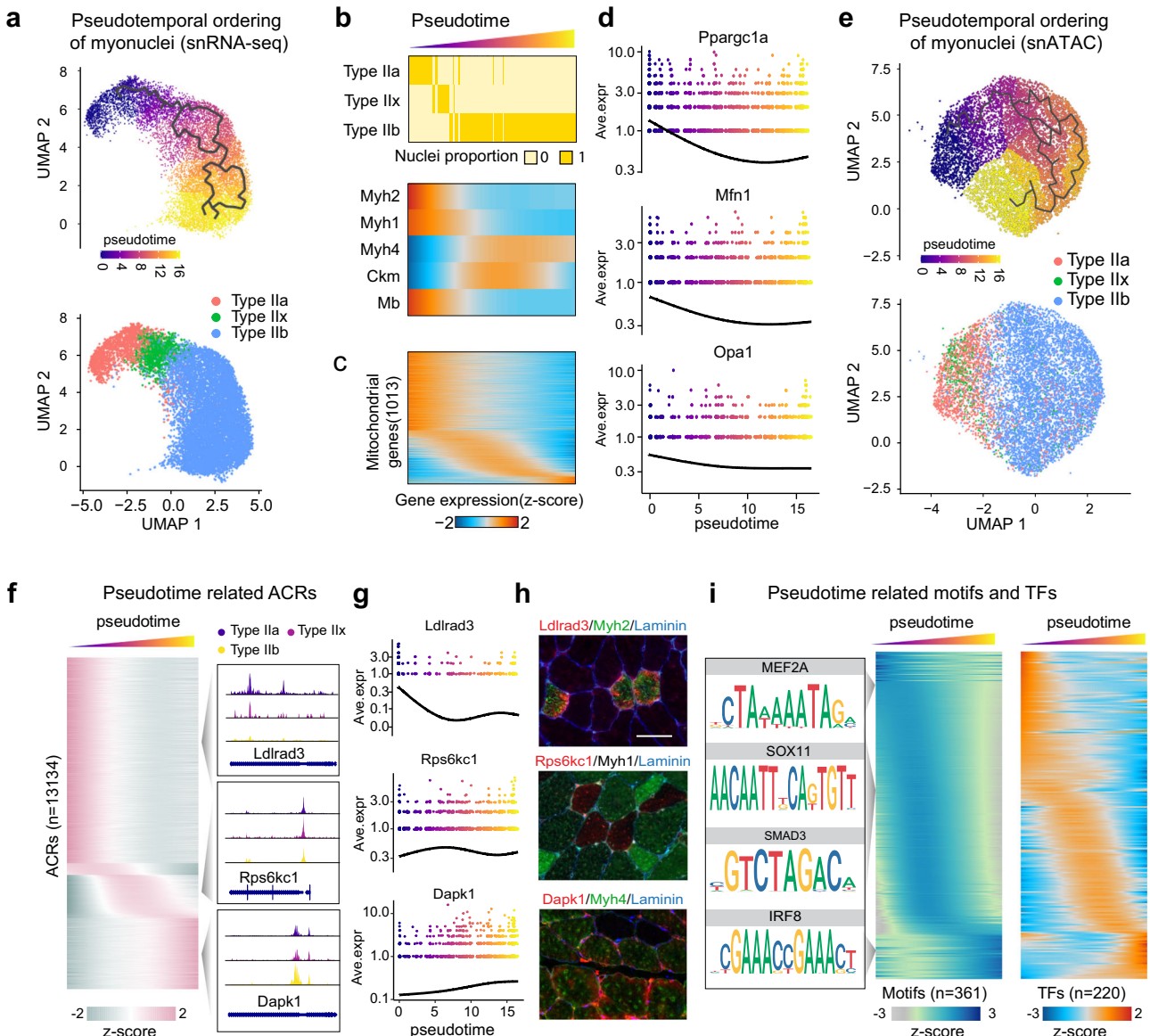

**Fig. 2 | Cis-regulatory dynamics underpinning myonuclear type transition.**
**a** UMAP representation of pseudotime trajectory in RNA expression for myonuclei (upper panel) and the corresponding myonuclei subtype alterations (lower panel). **b** Heatmap displaying the proportion of myonuclei of each nuclear type along the pseudotime progression. **c** Upper: Z-score heatmap of *Myh2, Myh1, Myh4, Ckm*, and *Mb* gene expressions over the myonuclear pseudotime trajectory. Lower: Heatmap of averaged expression of 1013 mitochondrial genes along the myonuclear pseudotime trajectory. **d** Gene expression dynamics of selected mitochondrial genes along the pseudotime trajectory of myonuclei. **e** UMAP visualization of chromatin accessibility highlighting the pseudotime trajectory of myonuclei (upper panel)

and alterations in myonuclei subtypes (lower panel). **f** Left: Z-score heatmap of 13134 pseudotime-related ACRs ($p < 0.001$). Right: Chromatin accessibility profiles of 3 selected genes along the myonuclear trajectory. **g** Gene expression dynamics of selected genes along the myonuclear trajectory. **h** Immunostaining of proteins encoded by the 3 selected genes (red) and myofiber marker Myh2 and Myh4 in GAS muscle of a normal mouse. Scale bar: 25 μm. **i** Relative motif activities of 361 TF motifs enriched in normal myonuclei over the pseudotime (middle), with corresponding TF expressions (right). Left panel shows the trajectory of four exemplified motifs.

a phospholipid flippase involved in phosphatidylserine and $Ca^{++}$ signaling[29] (Fig. 3k, right). GO enrichment analysis on the DAR-linked genes further indicated that DARs with reduced expression in denervated muscle are mainly enriched in the cation channel activities or ligand-gated cation channel activities. Conversely, DARs with increased expression upon denervation are predominantly enriched in terms for extracellular matrix structural constituent, integrin binding (Fig. 3l). Using the same approach, we likewise assessed DARs in adipocytes, MuSCs, endothelial cells, and macrophages, the results and conclusions of which are detailed in Supplementary Fig. 4c–h. In summary, our snATAc-seq captured the distinct changes of chromatin accessibility in each cell type of denervated muscle, which allows us to

characterize *cis*-regulation rewiring and to Identify muscle atrophy-related CREs in a cell-type-specific manner.

## Reprogramming of *cis*-regulation in denervated muscle
Given that TF signatures define cell types in normal muscle (Fig. 1), changes in TF activity should determine the phenotypes of atrophic muscle. To identify atrophy-responsive TFs, we performed motif enrichment analysis on cell-type-specific ACRs of normal and denervated muscle and visualized the active motif landscape changes induced by denervation (Fig. 4a). We identified the most differential motifs in myofibers (500), followed by macrophages (189 motifs), endothelial cells (86 motifs), and adipocytes (36 motifs). After

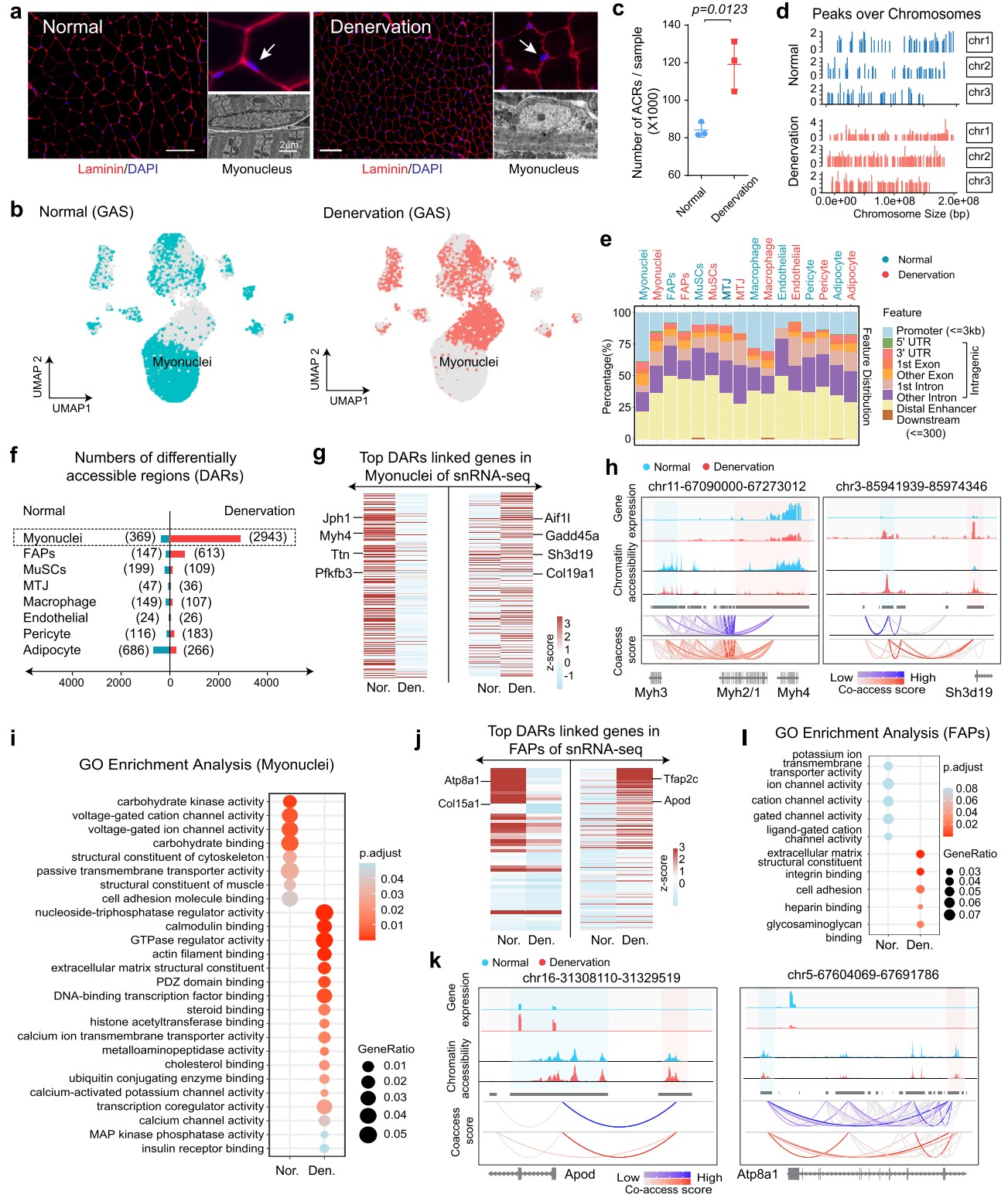

combining motif enrichment with the gene expression of corresponding TFs, we defined the atrophy-related motifs and their best-matching TFs in each cell type (Supplementary data 4). We then selected the top-ranking motif/TF pairs involved in myofiber atrophy (Fig. 4b), including TFs with no prior evidence of involvement in muscle atrophy. For example, *Tead4, Sox9,* and *Elk4* had motif activities and gene expression strongly induced in denervated myonuclei.

We further applied single-cell regulatory network inference and clustering (SCENIC) to our data, which method was designed to use

expressed genes to infer TFs and a set of genes directly controlled by them, termed a regulon[28]. By complementing motif enrichment-based TF inference, SCENIC allowed us to identify atrophy-related TFs and their target genes in our snRNA-seq datasets. We applied ChromVar analysis in our snATAC dataset to identify differentially enriched motifs and cognate TFs in denervated myonuclei. In parallel, we ran SCENIC analysis in our snRNA-seq dataset to predict TFs with their direct targets (regulons) in denervated myonuclei. To ensure each TF has both accessible motifs and co-expression of targeted genes, we

**Fig. 3 | Characterization of the cell-type-specific regulatory landscape in denervated muscle. a** Immunofluorescence staining of nuclei and chromatin in normal (left) and denervated (right) GAS muscle, with laminin in red and DAPI in blue. Scale bar: 50 µm. Electron microscopic images are provided for visualization of heterochromatin and nucleoli. **b** UMAP plot displaying distributions of chromatin accessibility in normal (left) and denervated (right) muscle. **c** Scatter-box plot illustrating the average number of detected regions in normal and denervated muscles. Data are presented as mean ± SEM ($n = 3$ mice/group), Significance was evaluated using a two-tailed unpaired t-test (t = 4.337, df = 4). **d** Fragment coverage plot displaying peak regions over chromosome 1-3 in normal (upper) and denervated (lower) muscle. **e** Bar plot of annotated the locations of differentially accessible regions (DARs) for each cell type. **f** Number of DARs between normal and denervated muscle categorized by cell type. **g** Heatmaps displaying normalized snRNA-seq expression of genes with altered expression linked to DARs between normal and denervated myonuclei. A complete list of these genes is provided in Supplementary data 3. **h** Fragment coverage tracks illustrating chromatin

accessibility and gene expression in normal versus denervated myonuclei. The purple (normal muscle) and red (denervated muscle) lines represent co-accessibility link scores for DARs, deeper in color indicating higher interaction levels. Pink-shaded regions highlight ACRs that overlap with gene promoter areas, while light blue-shaded regions pinpoint distal intergenic areas that show significant co-accessibility with promoter regions. **i** Dot plot depicting the comparison of Gene Ontology (GO) function enrichment between normal and denervated myonuclei. Dot color corresponds to p-values from the Benjamini-Hochberg-adjusted one-sided hypergeometric test, dot size represents the gene ratio of a GO term. **j** Heatmaps displaying expression levels of genes linked by co-accessibility to DARs between normal and denervated FAPs. **k** Fragment coverage tracks depicting chromatin accessibility and gene expression for chosen genes in normal and denervated FAPs. The color schemes align with those described in (**h**). **l** Comparison of GO function enrichment between normal and denervated FAPs. The color and size criteria for the dots are described in (**i**). Source data are provided as a Source data file.

intersected the ChromVar and SCENIC results, which yielded 125 mutual TFs (Supplementary data 5). We selected several top TFs to show in Fig. 4c and illustrate four TFs including *Elk4*, *Ar*, *Nr3c1*, and *Myog* as case studies showing the expression of TFs and their target genes in normal and denervated muscle (Fig. 4d, Supplementary Fig. 5a, b). Worth noting is the significant reduction in androgen receptor (*Ar*) and associated genes during post-denervation muscle atrophy, implying a potential role for *Ar*. Despite this, the lack of gender disparity in atrophy severity suggests the influence of *Ar* may not be decisive in denervation-induced muscle atrophy[30]. Further investigations are required to ascertain *Ar*'s role and uncover additional contributing mechanisms. All told, our findings highlight myofiber-specific TFs that are associated with muscle atrophy induced by denervation.

Since the activation of transcriptional enhancer regions is a key determinant of gene expression and phenotype emergence[31], we attempted to identify cell-type-specific candidate enhancers using our muscle CRE atlas. To this end, we performed H3K27ac ChIP-seq in normal and denervated muscle, H3K27ac being the mark of a distal enhancer[32]. At the global level, the H3K27ac landscape exhibited significant gain of signal in denervated muscle compared to normal, especially at distal enhancer regions (Fig. 4e, Supplementary Fig. 5c). By examining those distal DARs that have both co-accessibility with promoter regions and differential H3K27ac signal strength between normal and denervated muscle, we inferred candidate enhancers with increased or decreased H3K27ac signal in each cell type of denervated muscle (Fig. 4f, Supplementary data 6–7). To support our inference of candidate enhancers, we tested two candidates, one having strong H3K27ac signaling and co-accessibility with the promoter region of *Smox* in normal muscle and the other linked to *Gadd45a* and having increased H3K27ac signaling in denervated muscle. We cloned these candidate enhancers into report vectors and performed in vivo enhancer assays in TA (tibialis anterior) muscles with or without denervation (14 days). As shown in Fig. 4g, the enhancer of *Smox* had strong reporter signal in normal TA muscle, but its activity vanished after 14 days of denervation. In contrast, the enhancer of *Gadd45a* was quiescent in normal muscle but robustly activated by denervation (Fig. 4h). Consistent with these results, *Smox* mRNA was suppressed and *Gadd45a* mRNA significantly induced by denervation (Fig. 4g, h). These results provided experimental verification for our identification of enhancers in atrophic muscles. Thus, our muscle CRE atlas can be used to assign atrophy-related transcriptional regulators involved in atrophic muscle and other myopathies.

### The dynamics of chromatin accessibility during myofiber atrophy
Given that of the cell types examined, myonuclei undergo the most significant changes in transcriptional profile in response to

denervation, we performed a trajectory analysis in pooled normal and denervated myonuclei from the snRNA-seq dataset. All myonuclei formed a manifold continuum (Fig. 5a), which reflected a dynamic transitioning of gene regulatory networks. Indeed, the ACRs of normal and denervated myonuclei also exhibited a transition trajectory in pseudotime (Fig. 5b). We next aimed to interrogate chromatin dynamics along this trajectory. Evaluation of accessibility across pseudotime revealed a sharp increase in chromatin opening in denervated myonuclei (Fig. 5c), indicating that a large number of gene regulatory circuits are reactivated by denervation. For example, some genes that are silenced in mature myofibers, such as *Ttr* and *Myog*, showed an increase in the number and amplitude of peaks within their loci upon denervation, while analysis with the *Cicero* package also revealed re-connections of distal elements and proximal/intragenic elements (Fig. 5d). In denervated myonuclei, we also observed increased interactions between CREs in the locus of *Tnnt1*, an oxidative myofiber signature gene (Supplementary Fig. 6a), consistent with the finding that denervation stimulates the Type IIb to Type IIa/x myofiber transition[33]. However, full oxidative fiber properties are not acquired in this transition, because the chromatin accessibility and transcriptional levels of nuclear-encoded mitochondrial genes decline as denervation progresses (Supplementary Fig. 6b–g).

Using motif enrichment analysis, we further identified 579 TF motifs and 502 TF loci with differential chromatin accessibility across the transition trajectory (Fig. 5e). Among them, HINFP, NR3C1, MYOG, and ELK4 were identified as the most representative in each stage (Fig. 5e). Overall, there are three distinct dynamic patterns for these TFs (Fig. 5f): (1) Reduction of motif activity along the transition path; for example, the HINFP motif enrichment score was higher in normal type IIa/x myonuclei but decreased with prolonged denervation time. Target gene activity confirmed this dynamic. (2) Increasing motif activity at beginning of denervation followed by a decline as denervation progressed, exemplified by NR3C1. (3) De novo increases in motif activity and the accessibility of its cognate TF as denervation proceeds. For example, motif enrichment for MYOG began to increase persistently after 14 days of denervation, consistent with the increased activity of the cognate TF gene and predicted target genes as myofiber atrophy progressed. In summary, chromatin accessibility dynamics in myofibers reflect the genome-wide perturbations of gene regulation in denervated muscles.

### The hierarchy of atrophy-related TFs
Our SCENIC analysis of denervated myonuclei revealed that many TFs participate in a regulon driven by another TF, exemplified by *Tead4*, *Meis1*, and *Myog* being within the regulon of ELK4; this indicates a transcriptional hierarchy of *cis*-regulation in skeletal muscle. This observation spurred us to construct a myofiber-specific, atrophy-related TF regulatory network with consideration of the hierarchical

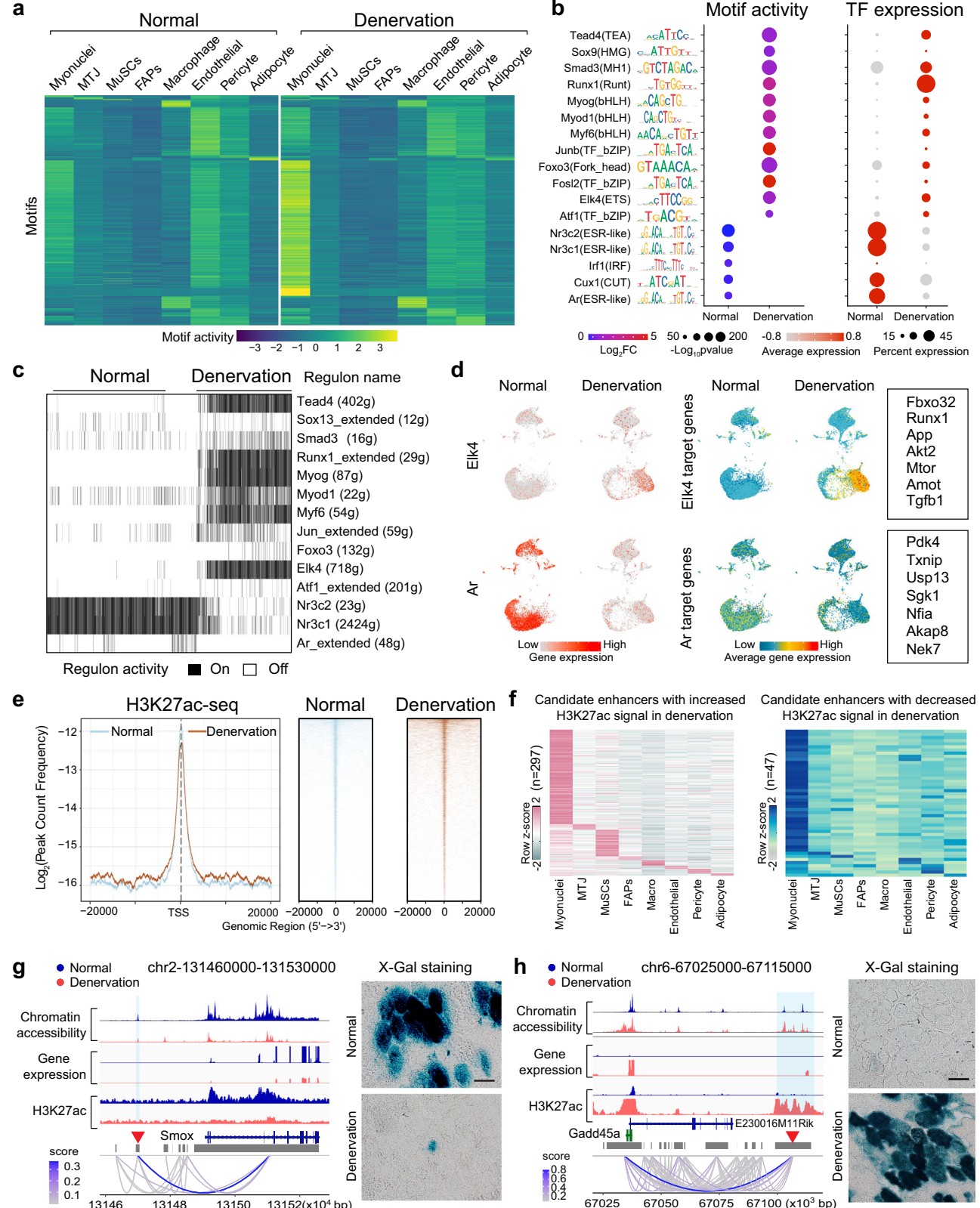

height of TF. Using the 125 mutual TFs yielded from ChromVar intersected with SCENIC (Supplementary data 5), we calculated the hierarchy height of each TF based on its incoming and outgoing edges, which resulted in three tiers of TFs[34,35]. By combining the hierarchy height and regulon connections identified for each TF, we built a regulatory network that exhibited a divergent pattern (Fig. 5g, Supplementary data 8), consistent with our earlier observation of

increased global ACRs during muscle atrophy. TFs of higher tier usually had amplified effects due to regulating the expression of more TFs and downstream target genes. Most high-tier TFs also exhibited de novo increased dynamics after denervation. For example, *Elk4*, which is of the highest tier, exhibited consistently increased motif activity and expression during denervation. To characterize the effects of this TF network in controlling gene expression in atrophic myofibers, we

**Fig. 4 | *Cis*-regulation rewiring in denervated muscle. a** Heatmap (Z-score) illustrates average chromVAR motif activity for each cell type in both normal and denervated muscles. Refer to Supplementary data 4 for atrophy-related motifs and their best-matched TFs specific to each cell type. **b** Dot plots display the differential motif activities (left) and corresponding TF expression (right) in normal and denervated myonuclei. Significance was evaluated with a Wilcoxon Rank Sum two-side test, *n* = 3 mice per group containing 14,388 nuclei in a joint analysis. **c** Heatmap presents select myonuclei-specific regulon activities in normal and denervated muscles, as identified by intersecting ChromVar and SCENIC results. The comprehensive list of TFs (regulons) is available in Supplementary data 5. The SCENIC algorithm binarized regulon activity as "On" (black) or "Off" (white). **d** UMAP plots showcase the upregulation of Elk4 and its target genes (upper), or the downregulation of Ar and its target genes (lower) in response to denervation. **e** H3K27ac ChIP-seq profile of normal and denervated muscles. Left: the read count frequency in selected range around TSS. Right: heatmaps of normalized H3K27ac tag densities at differentially H3K27-acetylated regions. **f** Heatmap displays candidate enhancers with either increased (left) or decreased (right) H3K27ac signals in denervated muscles. For a complete list of inferred candidate enhancers, refer to Supplementary data 6 and 7. **g, h** Left: Fragment coverage tracks showing chromatin accessibility, gene expression, and H3K27ac ChIP-seq signals. Red arrowhead indicates the tested regions of enhancers. Right: Representative image of in vivo enhancer activity visualized by X-Gal staining in tibialis anterior (TA) muscles from 3 normal and 3 denervated mice. Scale bar: 50 μm. Source data are provided as a Source data file.

compared the expression and functional composition of downstream-targeted genes according to their GO term annotations (Fig. 5g). For example, catabolic effectors such as ubiquitin conjugating enzyme and E3 ligases were controlled by multiple TFs and induced by denervation. Interestingly, we also observed many TFs with increased activities to be linked to reduced target gene expression, exemplified by energy metabolism genes; these TFs might act as transcriptional repressors in the regulatory program of atrophic myofibers. Taken together, we successfully constructed a hierarchical regulatory network of TFs that integrated expression and the connectivity features of atrophy-related genes in skeletal muscle.

## ELK4 regulates muscle protein loss in denervated muscles

From our hierarchical atlas of atrophy-related TFs, we recognized that ELK4, an ETS domain-containing TF, occupies the highest tier, implying that it could be a hub in regulating muscle atrophy. snRNA-seq revealed that *Elk4* is predominately expressed in normal type IIb myonuclei and is strongly up-regulated by denervation. These results were supported by increased *Elk4* gene body accessibility, transcript abundance in bulk RNA-seq, and H3K27ac signal within the *Elk4* locus (Fig. 6a). We next estimated the accessibility dynamics of *Elk4* along the trajectory from normal to denervated myonuclei from our snATAC-seq data and found *Elk4* gene activity to increase de novo as denervation progressed (Fig. 6b). This pattern was corroborated by real-time qPCR and immunostaining, which indicated a progressive increase in ELK4 expression with time of denervation (Fig. 6c, d, Supplementary Fig. 7a). Using Signac footprinting (TF occupancy) analysis, we visualized a well-defined footprint surrounding ELK4 motifs and determined that ELK4 motif enrichment increased at the genome-wide level in denervated myonuclei (Fig. 6e). This indicates that denervation stimulates the physical interaction between ELK4 and its target genes.

Unsupervised regulator analysis further inferred a set of genes directly regulated by ELK4, including *Tgfb1*, *Cpne2*, and *Chrd* (Fig. 6f). GO enrichment analysis of this regulon highlighted the processes of skeletal muscle development and protein catabolism (Supplementary Fig. 7b), suggesting that ELK4 could act as a catabolic TF during muscle atrophy. To validate this prediction, we force-expressed full-length, untagged mouse *Elk4* in $C_2C_{12}$ myocytes and performed total RNA-seq; the resulting data revealed extensive alterations in gene expression induced by *Elk4* overexpression, including induction of *Tgfb1* and other inferred target genes in the ELK4 regulon (Fig. 6g). We then focused on the regulation of *Tgfb1* because it plays an apparent role in muscle catabolism[36]. Homer motif analysis identified four ELK4 binding sites within the promoter region of the *Tgfb1* locus (Fig. 6h). Using RT-qPCR and immunoblotting, we confirmed the activation of TGFB1 signaling in *Elk4*-overexpressing $C_2C_{12}$ myocytes and denervated GAS muscle (Fig. 6i, j, Supplementary Fig. 7c), consistent with the conclusion that TGFB1 exerts its catabolic role in an autocrine or paracrine mode[37]. Given that denervation increases intercellular $Ca^{++}$ concentration in myofibers[23], we mimicked this response by treating $C_2C_{12}$ myotubes with ionomycin, which induces a high level of intercellular

$Ca^{++}$ in many cell types[38]. Ionomycin significantly increased the level of *Elk4*, and this response was associated with both induction of *Tgfb1* and remarkable myotube atrophy. Conversely, silencing of *Elk4* prevented ionomycin-induced *Tgfb1* expression while partially reversing myotube atrophy and atrophy-associated genes (Fig. 6k–m). These results not only confirmed that ELK4 regulates the expression of *Tgfb1* but also suggest ELK4 to act as a $Ca^{++}$-responsive TF.

To examine whether inhibition of ELK4 could limit myofiber atrophy in denervated muscles, we electroporated the TA muscle of mice with either a triad *Elk4* shRNA plasmid (pAAV-shElk4-GFP) or control plasmid (pAAV-scrambled shRNA-GFP) immediately after denervation. After two weeks, gene delivery efficiency was evaluated by the expression of GFP intensity (Fig. 7a, Supplementary Fig. 7d). ELK4 protein level was found to be reduced ~85% compared to mice electroporated with control plasmid (Fig. 7e). In mice with denervation, there was a 40% decrease in TA muscle weight after transfection with the scrambled shRNA; however, transfection with the triad *Elk4* shRNA attenuated this decline by 42% (Fig. 7a, middle panel). That this response was elicited by *Elk4* inhibition was evident in the ipsilateral, non-transfected GAS muscles being similarly reduced by denervation (Fig. 7a, right panel). Size distribution analysis and average CSA (Cross-sectional area) performed on GFP-positive myofibers showed an obvious rightward shifting of CSA in denervated TA muscle electroporated with *shElk4* versus controls, consistent with the results of average CSA (Fig. 7c, d). In addition, expression of TGFB1 and phosphorylation of SMAD3 were significantly decreased in denervated TA muscles with *Elk4* knockdown (Fig. 7e). Moreover, we detected that expression of atrophy-related genes including *Atrogin-1(Fbxo32)*, *MuRF1(Trim63)*, and other ELK4 target genes was significantly decreased by *Elk4* knockdown in denervated TA muscles (Fig. 7f). Thus, we concluded that *Elk4* inhibition alleviates the muscle wasting caused by denervation via suppression of TGFB1-related catabolic signaling.

## Discussion

Over the past few decades, much effort has been devoted to studying how muscle-cell-sensed signals are transduced into the nucleus and cause phenotypic changes, with only limited understanding of how these signals lead to chromatin changes and *cis*-regulatory remodeling. However, a comprehensive understanding of CREs and TFs in myofibers and muscle-resident cells and their dynamic interactions in response to environmental changes is critical to unraveling the determinants of cellular identity and the sources of changes in gene regulation. With this in mind, we integrated several unbiased high-throughput approaches to construct an inclusive roadmap of DNA opening and *cis*-regulatory signatures in specific muscle cell types along with their trajectories from normal states to catabolic ends. These data not only fill gaps in our knowledge but, from an applied perspective, they pave the way for development of future gene editing applications or gene therapy methods by which to combat muscle diseases. For example, our results suggest that cellular identity in skeletal muscle is established by combinatorial TF activity

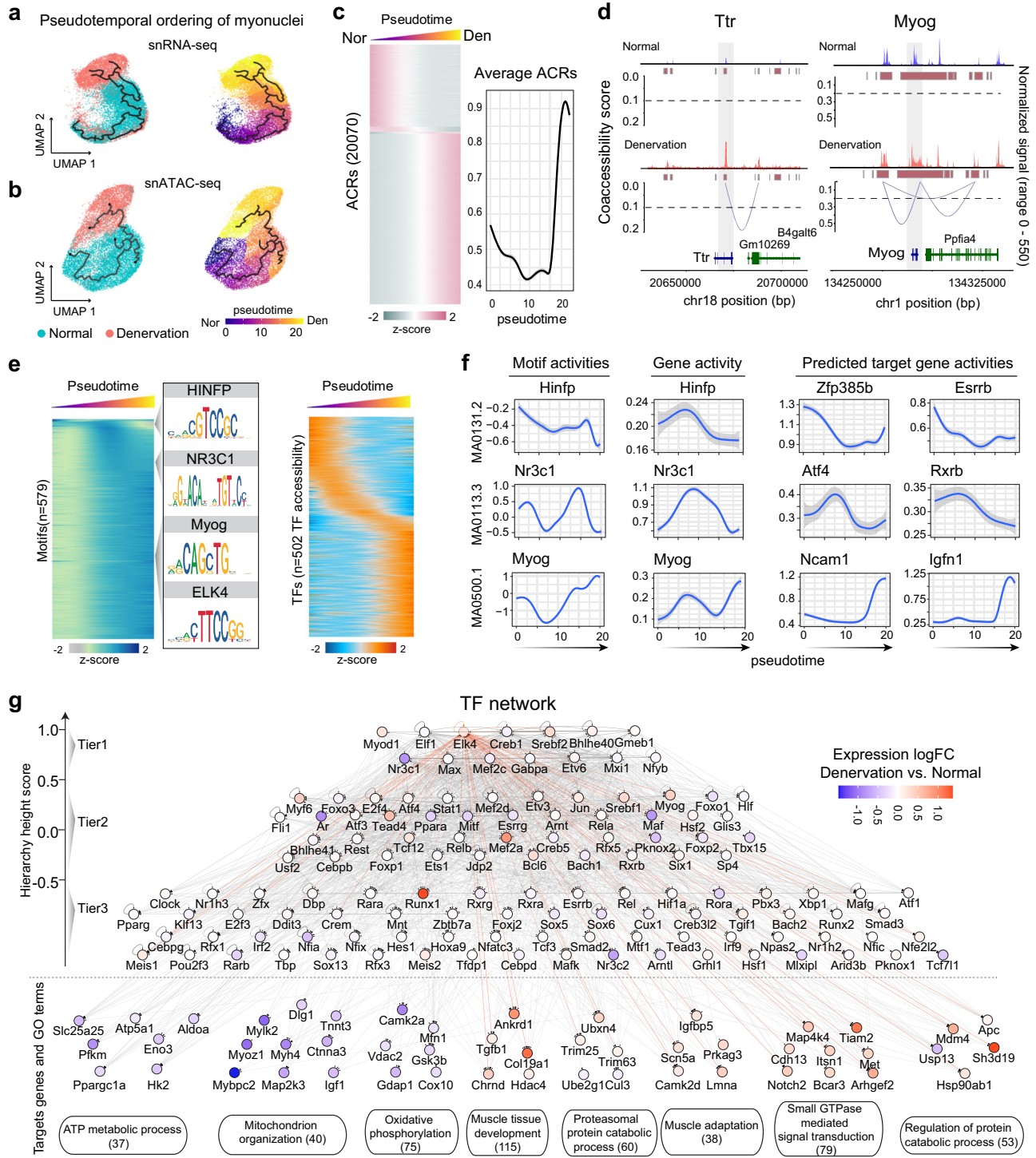

**Fig. 5 | Dynamic chromatin accessibility and TF network in denervated myonuclei. a** UMAP of mRNA profile illustrating transition trajectory from normal to denervated myonuclei (left) and pseudotime progression (right). Nor: normal, Den: denervation. **b** UMAP of ACRs profile depicting transition trajectory and pseudotime from normal to denervated myonuclei. **c** Left heatmap showing relative accessibilities of 20,070 pseudotime depended ACRs, average of ACRs accessibility across pseudotime is shown on the right ($p < 0.01$, significance was evaluated with a Benjamini-Hochberg-adjusted regression analysis). **d** Fragment coverage tracks showing chromatin accessibility and co-accessibility links in normal and denervated myonuclei. The region of increased accessibility in denervated muscle is

shaded in gray. **e** Relative motif activities for 579 TF motifs (left) and their corresponding TFs' expression (right) across pseudotime. Four examples of enriched motifs are shown in the middle. **f** Line plots illustrating the dynamic changes of chromatin accessibility in selected in TFs (left), motif activity (middle), and activities of target genes (right) from GAS muscles of 3 normal and 3 denervated mice. Shadow represents the 95% confidence interval of local polynomial regression fitting. **g** Gene regulatory network with TF hierarchy in denervated muscle. TFs are stratified into three tiers by hierarchy height score. Target genes are grouped by terms of GO enrichment. For detailed hierarchy height and GO terms, see Supplementary data 8.

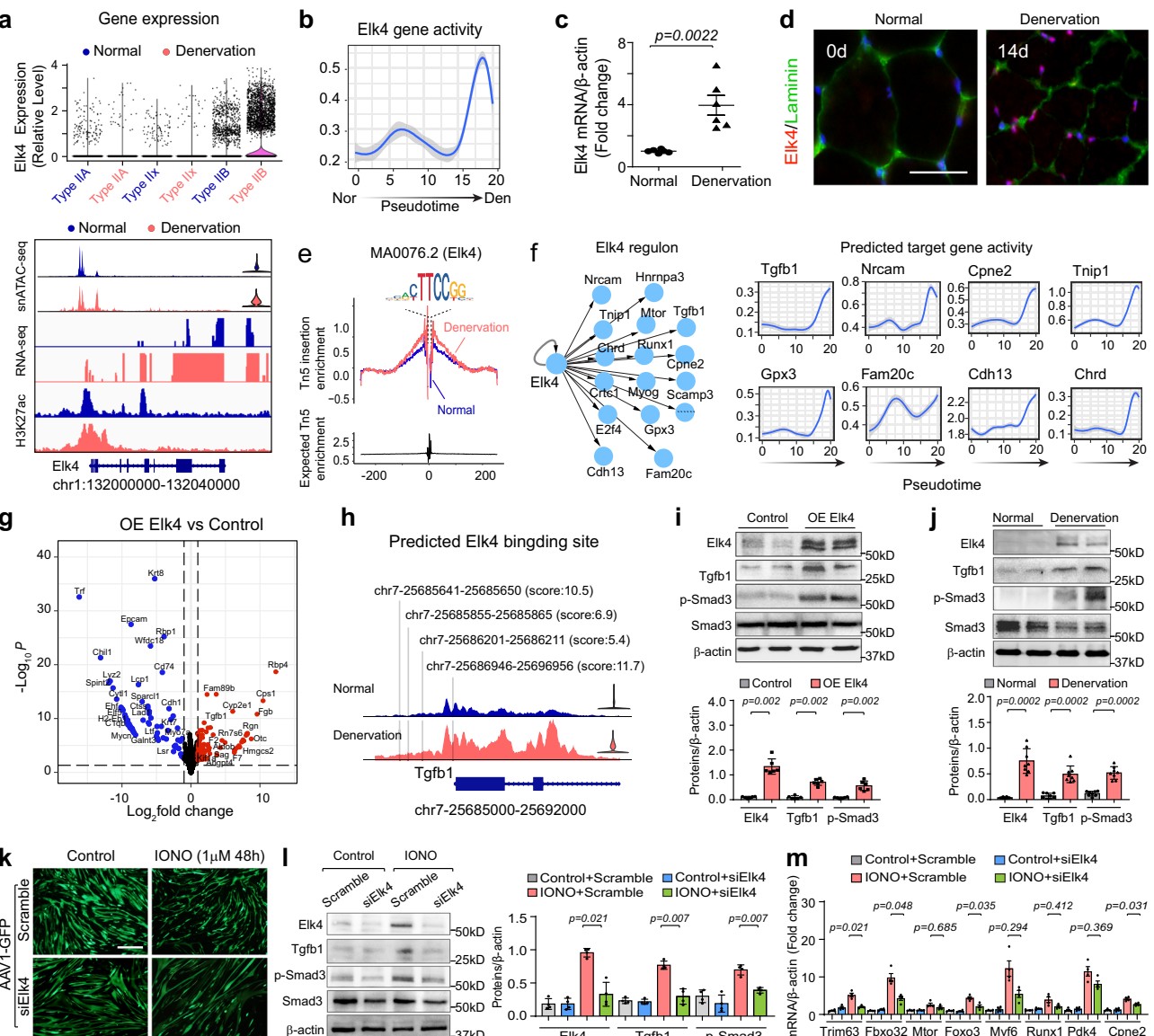

**Fig. 6 | Elk4 regulates myotube atrophy through TGFB1. a** Upper: Violin plot of Elk4 expression across myonuclei types. Lower: Tracks show chromatin accessibility, Elk4 expression, and H3K27ac profile at the Elk4 locus. **b** Line plots track Elk4 activity dynamics from normal to denervation. The solid line represents the Local Polynomial Regression fit, with the shaded region indicating the 95% confidence interval. **c** Quantitative RT-PCR analysis of Elk4 mRNA in normal and denervated GAS muscle (n = 6). **d** Immunofluorescence staining of Elk4 in normal and denervated GAS muscle. Scale bar: 25 µm. **e** TF footprinting plot demonstrating Elk4 motif activity in normal and denervated myonuclei. **f** Transcriptional network map of Elk4 regulons (left) and line plots of gene activity dynamics for Elk4 target genes across pseudotime. **g** Volcano plot of transcriptomic changes in C2C12 cells post Elk4 overexpression. Red and blue dots signify up- and down-regulated genes with log2-fold change thresholds of ±1 and *p*-value < 0.05, analyzed using a two-sided Fisher's exact test in EdgeR v3.12.1. **h** Fragment coverage plot showing Elk4 motif within the promoter region (2000bp from TSS) of the Tgfb1 locus. Predicted scores

are shown in parentheses. **i** Immunoblotting and quantitative analysis of ELK4, TGFB1, and p-SMAD3 in C2C12 cells with or without Elk4 overexpression (n = 6). **j** Immunoblotting and quantitation of ELK4, TGFB1, and p-SMAD3 in normal and denervated GAS muscle (n = 8). **k** Fluorescence images displaying changes in C2C12 myotubes (transfected with AAV1-Scramble-GFP or Elk4 siRNA-GFP) in response to ionomycin (IONO, 1 mM, 48 h). Scale bar: 50 µm. **l** Immunoblotting (left) and quantitative analysis (right) of ELK4, TGFB1, and p-SMAD3 for experiments in (**k**) (n = 4). **m** Quantitative RT-PCR analysis of atrophy-related genes in normal or ionomycin-treated C2C12 myotubes with or without Elk4 knockdown (n = 4). All bar graphs present quantitative data as mean ± SEM. "*n*" denotes the number of biological replications (**i, l, m**) or mice/group (**c, j**). Significance was assessed using a two-side Wilcoxon rank-sum test (**c, i, j**) or two-side one-way ANOVA with post hoc Tamhane's multiple comparisons test (**l, m**). Source data are provided as a Source data file.

and the accessibility of corresponding target binding sites, implying that reprogramming of these interactions in muscle cells can reverse muscle atrophy, a long-held goal. In addition, sequences associated with CRE activity can be used to tailor the cell-type specificity of transgene expression cassettes or to direct endogenous gene silencing. Moreover, cell-type-specific CREs and co-accessible ACRs provide a high-confidence list of sequences that can be used in genome editing to alter interactions between enhancers and promoters or

intergenic regulatory elements, resulting in improved muscle function and metabolism. Because some transcriptional responses triggered by denervation are similar to those triggered by other catabolic stimuli such as CKD or cancer, we also expect to see a similar organization of regulatory networks in other muscle atrophy-related stresses. Our analysis of *cis*-regulatory dynamics from the catabolic perspective thus has implications for future experiments in other myopathies.

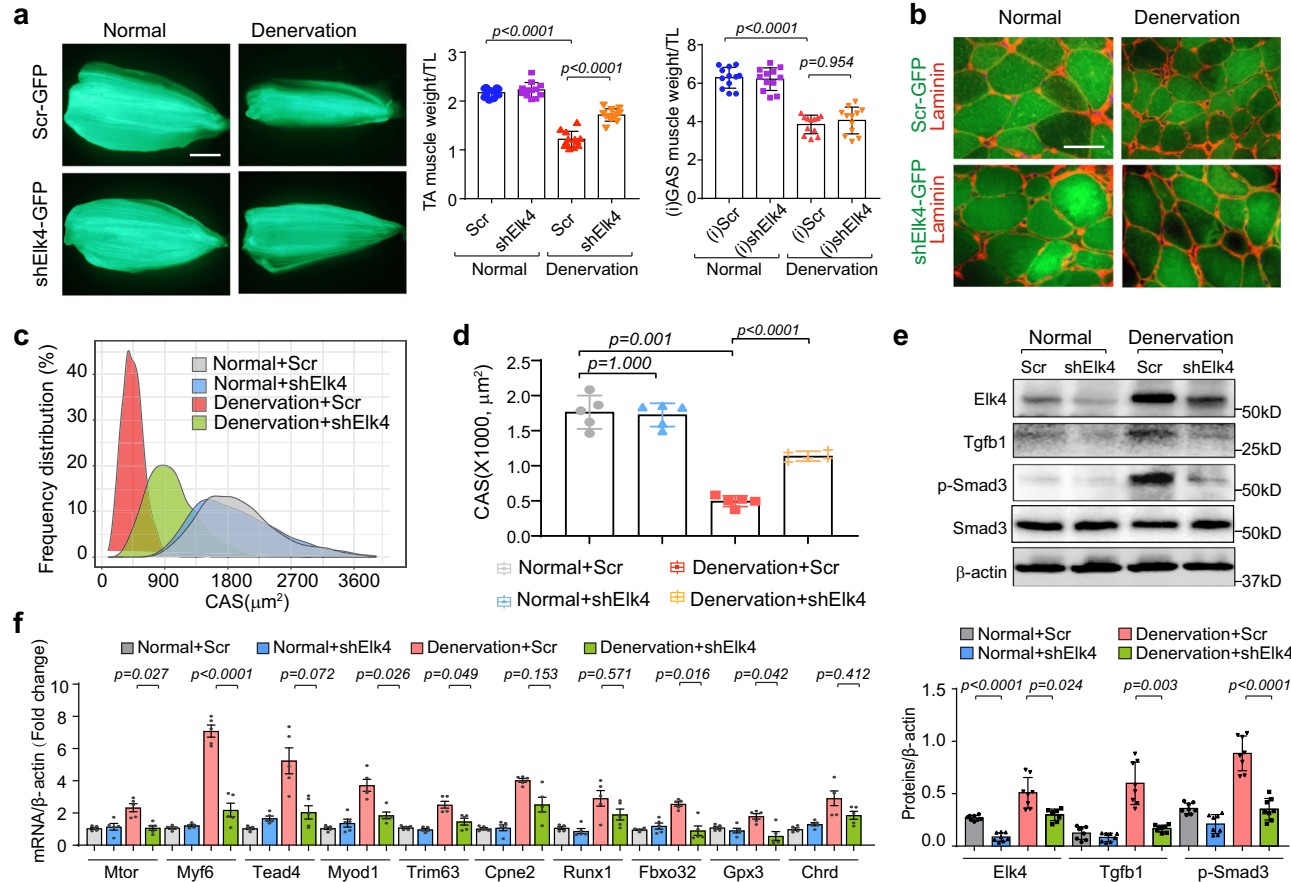

**Fig. 7 | Elk4 inhibition attenuates muscle atrophy induced by denervation.**
**a** Left: Fluorescence images of TA muscles transfected with pAAV-GFP plasmid containing either scramble (Scr) shRNA or a combination of three pAAV-GFP plasmids each with a specific Elk4-targeting shRNA. Middle: Scatter-bar plot comparing the weight of transfected TA muscles. Right: Scatter-bar plot of the weight of ipsilateral (non-transfected) GAS muscles 14 days post-denervation. Data are presented as mean ± SEM (*n* = 12 mice/group). Significance was determined by two-sided one-way ANOVA with post hoc Tamhane's multiple comparisons test. Scale bar: 2 mm. **b** Cross-sectional images corresponding to samples in (**a**). "CSA" indicates the cross-sectional area. Scale bar: 25 μm. **c** Distribution of myofiber sizes derived from the samples in (**b**). **d** Quantitative analysis of the cross-sectional area

based on the samples from (**b**). Data are presented as mean ± SEM (*n* = 5 mice/group, significance was determined by two-sided one-way ANOVA with post hoc Tamhane's multiple comparisons test. **e** Western blot analysis (left) and its quantification (right) using samples from (**a**). Data are shown as mean ± SEM (*n* = 8 mice/group, significance was determined by two-sided one-way ANOVA with post hoc Tamhane's multiple comparisons test). **f** Quantitative RT-PCR analysis of atrophy-related genes in TA muscle, comparing normal versus denervated conditions with or without Elk4 knockdown. Data are presented as mean ± SEM (*n* = 5 mice/group, significance was determined by two-side one-way ANOVA with post hoc Tamhane's multiple comparisons test). Source data are provided as a Source data file.

Early on, comparative gene expression profiles of different atrophy models led to the discovery of several genes that are coordinately regulated by distinct TFs. Since then, studies of transcriptional regulation have generally focused on these master regulators, which include FOXOS, NF-κB, RUNXS, and MYOG. Although these discoveries established the concept that transcription factors drive muscle atrophy, systematic characterization of TFs and their interacting CREs remains challenging because CREs can exert their effects through long-range chromatin interactions. Therefore, genome-wide assays (such as single-cell omics and epigenomic profiling) and bioinformatics are needed to decipher these regulatory networks. By leveraging these techniques, we confirmed these master TFs as major contributors to the regulation of gene expression in atrophic muscle. Furthermore, we found that not only the master regulators are required to achieve catabolic-stimulated transcriptome changes, with many atrophy-responsive genes, such as genes involved in protein degradation and energy metabolism, being dynamically targeted by multiple transcription factors. Thus, muscle atrophy responses should be viewed as orchestrated by master regulators and facilitated by other TFs that are coordinated by signaling pathways, ultimately leading to the initiation of transcriptome changes.

One important finding of this study is that denervation enhances the genome-wide acetylation of chromatin as detected by H3K27ac-ChIP sequencing. This result is consistent with a previous observation that acetylation of histones is increased in denervated myonuclei[39]. However, we and others also observed that muscle denervation is accompanied by induction of class IIa histone deacetylases (HDACs) activity and HDAC-dependent gene expression[40]. These results might point to a transcriptional feedback strategy in the response to neurogenic muscle atrophy, presumably to allow rapid restoration of innervation and neural activities once the stress is lifted, given that HDACs control the expression of myogenin and synaptic genes[41]. Apparently, these feedback responses redeploy embryonic or developmental programs in atrophic muscle, culminating in perturbations of adult gene regulatory networks in adulthood, especially when HDACs are associated with the expression of atrophy-related genes that continue to fuel catabolic pathways and accelerate muscle protein breakdown before the catabolic stress is eliminated. Since some of the transcriptional responses in denervated muscle resemble those elicited by other catabolic stimuli, we would expect to see similar chromatin reorganization and feedback networks in other stress-induced muscle atrophy.

There is clear evidence that TGF-β signaling is one of the major mechanisms leading to loss of muscle mass and function. Both TGF-β1 and myostatin exert catabolic effects through autocrine signaling[42]. In denervation-induced muscle atrophy, TGF-β1, but not myostatin, is induced and is responsible for activating SMAD2/SMAD3 downstream signaling. However, there are few reports on how the expression of TGF-β1 is regulated. In this study, we provided multiple layers of evidence, from the opening of the *Tgfb1* genomic locus to the expression of TGF-β1 target genes, demonstrating that ELK4 promotes TGF-β1 expression in denervated muscle. Given the numerous biological effects of TGF-β1 in regulating organ fibrosis, myogenesis, and tumor metastasis, our discovery of this ELK4-TGF-β1 axis could have broad implications and possibly enable new strategies to combat the anomalous responses induced by TGF-β1.

Although we provide evidence such as histological analysis and bulk assays (total RNA-seq and ChIP-seq) to support chromatin accessibility as a robust proxy of gene expression, per-nucleus sequencing depth and dropouts remain common technical limitations that influence the chromatin accessibility readout and cause omissions from our *cis*-regulatory atlas. In addition, muscle fibers are composed of both oxidative and glycolytic myofibers, each of which is a syncytium possessing a sophisticated arrangement of "nuclear domains"; single-cell assays disrupt this architecture and spatial localization. This common limitation makes it impossible for cell trajectory inference to distinguish whether myonuclear transition occurs preferentially in certain myofibers or uniformly across all myofibers. Therefore, our model of myofiber transition should be considered together with the spatial transcriptome of skeletal muscle[43]. As another limitation, it is well known that TF activity is also influenced by post-translational modifications; however, our TF network (Fig. 5g) cannot include the impacts of these modifications. Furthermore, the TF hierarchy was determined by computer algorithms based on presence of input and output connections, which cannot fully reflect the contribution of TFs to muscle catabolism. Thus, the TF network atlas should be considered preliminary, with potentially considerable ambiguity for further correction or validation and inference.

In summary, our results provide a rich resource for defining key regulatory events in skeletal muscle energy and protein metabolism under both healthy and pathological conditions.

## Methods

### Study animals
Animal procedures were approved by the Baylor College of Medicine Institutional Animal Care and Use Committee (AN-3965) or by the Ethics Committee of the Third Affiliated Hospital of Sun Yat-sen University (SYXK 2019-0136). Male C57BL/6J (12-week-old) mice were fed ad libitum on a standard laboratory diet, maintained under a 12-h light/dark cycle conditions. Denervation surgery were performed using well-established method. Briefly, after mice were anesthetized with 5% isoflurane, the left sciatic nerve near the femoral head was exposed and a 1.5-mm piece of sciatic nerve was removed. Gastrocnemius (GAS) or Tibialis anterior (TA) muscle samples were harvested at 14 days after denervation. GAS and TA muscles from left hind limb of normal mice served as normal controls[44].

Electroporation was performed at 100 volt for 10 pulses (20 ms each pulse) after injecting 60 μg of plasmid vector in 20 μl PBS into tibialis anterior (TA) muscle[45,46]. At 14-day after electroporation, denervated mice or/and normal mice were anesthetized, and TA and GAS muscles were collected for further examinations. The untagged or green-fluorescent-protein (GFP) tagged mouse Elk4 (NM_001376954.1) pAAV expression vectors (VB900124-9241zuq, VB900138-4352whw), GFP marked Elk4 pAAV-shRNA knockdown vectors (VB900138-4353mwt, VB900129-1910enq, VB900138-4354tjw), and related control vectors (VB010000-9394npt, VB010000-0023jze) were purchased form VectorBuilder (Chicago, IL, USA).

### Muscle single-nuclei isolation
Nuclei extraction protocol was adapted from Krishnaswami et al. with modifications[7]. Briefly, homogenization buffer was prepared with Nuclei Isolation Media1 (NIM1: 320 mM sucrose, 25 mM KCl, 5 mM MgCl₂, and 10 mM Tris buffer, pH = 8.0) containing 1 μM DTT, 0.4 U/μL RNase Inhibitor (NEB Inc, Ipswich, MA), 0.20 U/μL SUPERase-In RNase Inhibitor (Thermo Fisher Scientific Inc, Waltham, MA), and 0.1% Triton X-100. 100–150 mg GAS muscles were minced on ice with a razor blade and then suspended in 5 mL ice-cold homogenization buffer and homogenized with a dounce grinder for 16 strokes on ice. The homogenate was sequentially filtered through 70 μm, 40 μm and 25 μm cell strainers (pluriSelect, El Cajon, CA) and centrifuged (1000 × g) for 5 min at 4 °C to pellet the nuclei. The nuclei were resuspended with 1 mL of ice-cold wash buffer (PBS containing 2% bovine serum albumin and 0.2 U/μL RNase inhibitor) followed by filtering through the 15 μm cell strainer before centrifugation (500 × g) for 5 min at 4 °C.

### Single-nucleus RNA-seq using 10x genomics chromium
Six snRNA-seq libraries were prepared (GSE183802)[7]. Briefly, resuspend nuclei in cold wash buffer for adjusting the concentration of nuclei to 1200 nuclei/μl with a hemocytometer. About 10,000 nuclei were loaded to each channel and then partitioned into Gel Beads in emulsion in the Chromium instrument using the Single-Cell 3′ Reagent Kit v3 according to the manufacturer's protocol (10× Genomics). Library preparation including reverse transcription, barcoding, cDNA amplification, and purification was performed according to Chromium 10x v3 protocols. Libraries were sequenced on the Illumina HiSeq 4000 platform using a custom paired-end sequencing mode[47]. A minimum of 50,000 reads per nucleus were sequenced and counted with Cell Ranger Single-Cell Software Suite 3.0.2 by 10x Genomics using a custom pre-mRNA GTF built on GRCm38 to include intronic reads.

### Single nucleus RNA sequencing bioinformatics processing
Analysis for single nucleus RNA sequencing is detailed in our previous published study[7,47]. Briefly, raw reads were mapped to the reference genome using STAR (Spliced Transcripts Alignment to a Reference) with default setting[48]. The mapped reads with valid barcodes and unique molecular identifiers (UMIs) were obtained. A quality control step was performed by removing doublets (DoubletFinder v2.0) and low-quality nuclei (less than 200 genes or more than 0.8% mitochondria genes) (Seurat v3.1.0) to ensure only transcripts originating from nuclei were retained. Genes detected in less than 3 nuclei were also removed. After quality control in Seurat package[49], the resulting expression matrix of each sample was normalized using the "NormalizeData" function. Then, the "IntegrateData" function was applied to integrate datasets and "ScaleData" function was used to scale the data.

**Dimension reduction and clustering.** The top 2000 variable genes were used for downstream principal component analysis (PCA) prior to dimensionality reduction. Top 10 principal components were then input for Uniform Manifold Approximation and Projection (UMAP) dimensionality reduction. Clusters were identified using FindCluster function (resolution = 0.5). The cluster annotation was based on the expression of established cell type identities and refined using the signature genes identified by FindAllMarkers function (Wilcoxon Rank Sum test, min.pct = 0.25, logFC.threshold = 0.25).

**Trajectory analysis.** Trajectory analysis was performed using the Monocle3 (version 0.2.3) algorithm with default parameters. The normalized counts matrix for the whole dataset or each cell/nucleus type were input to create CDS objects, followed by data normalization and PCA analysis, dimension reduction and cell clustering (cluster_cells, resolution = 0.0001). After the principal graph was plotted, the nulcei were ordered in pseudotime. All the trajectory graphs were visualized[50].

## Single-nucleus ATAC sequencing using chromium single-cell ATAC

After the isolation of single-nuclei, the nuclear pellet was re-suspended in Nuclei Buffer (10 × Genomics, Chromium Single-Cell ATAC kit). Nuclei suspension was loaded and incubated in transposition mix from Chromium Single Cell ATAC Library & Gel Bead Kit (10X Genomics, PN-1000110) by targeting 10,000 nuclei per sample. Libraries generated in each sample using the Chromium Single Cell ATAC Library & Gel Bead Kit and Chromium i7 Multiplex Kit N (10X Genomics, PN-1000084) were sequenced on Illumina HiSeq 4000 platform using 2 × 50 bp paired-end sequencing mode according to the manufacturer's protocol. Libraries were aggregated with Cellranger-atac (v1.2) without depth normalization. A mean of 214,400,536 reads were sequenced for each snATAC library corresponding to a mean of 15,285 fragments per nucleus.

## Single-nucleus ATAC-seq analysis

**Data processing and quality control.** All reads were mapped to the Mus musculus genome reference (mm10) using Cellranger-atac in accordance with 10x Genomics recommendations. Next, datasets were processed with Seurat v4.2.0 and Signac v1.8.0[49]. Low-quality cells were removed from the aggregated snATAC-seq library (peak region fragments >3000, peak region fragments <30,000, %reads in peaks >15, blacklist ratio <0.025, nucleosome signal <4 & TSS enrichment >2) before normalization with term-frequency inverse-document-frequency (TFIDF). After filtering, there was a mean of $4400 \pm 1321$ nuclei per snATAC-seq library with a mean of $9335 \pm 5669$ peak-region-fragments detected per nucleus.

**Dimension reduction, batch effect correction, and clustering.** Dimensional reduction was performed via singular value decomposition (SVD) of the TFIDF matrix and UMAP. A KNN graph was constructed to cluster cells with the Louvain algorithm. Batch effect was corrected with Harmony package using the "RunHarmony" function in Seurat on the 2 to 40 latent semantic indexing (LSI) components[51]. A gene activity matrix was constructed by counting ATAC peaks within the gene body and 2 kb upstream of the transcriptional start site (TSS) using protein-coding genes annotated in the Ensembl database. Data corrected by Harmony were used for unsupervised clustering analysis using "FindNeighbor" and "FindClusters" functions in Seurat[49].

**Nuclei type annotation.** Nuclei from the matched snRNA-seq analysis were used as a reference dataset to predict nucleus types in the snATAC-seq. This prediction used top 5000 variable features of the snRNA-seq data as reference, and the gene activity matrix generated using Seurat's "GeneActivity" from snATAC-Seq data as the query data. Transfer anchors were learned using "FindTransferAnchors". Then, anchor pairs were used to assign RNA-seq labels to the snATAC-seq nuclei type using "TransferData" with the snATAC-seq LSI reduction as weight. The result assigned each nucleus in the scATAC-seq with a cell type identity from the matching snRNA-seq data. Final nuclei annotation was based on both the prediction score and lineage-specific gene activity.

**Differentially accessible regions (DARs) analysis.** Differential chromatin accessibility between nucleus types (or between normal and denervation muscle) were assessed with the Signac "FindMarkers" function for peaks detected using a likelihood ratio test and a log2-fold-change threshold of 0.25. Bonferroni-adjusted p-values were used to determine significance at $p < 0.05$. Genomic regions containing snATAC-seq peaks were annotated with ChIPSeeker (v1.28.3) and ClusterProfiler (v4.0.5) using the UCSC database on mm10[25,52,53].

**Motif enrichment analysis.** To discover transcription factor (TF) dynamics and variation in their motif accessibility, we conducted motif analysis using ChromVar (v1.14.0)[15]. The positional weight matrices (PWMs) for 579 known TFs were obtained from the JASPAR2018 database. Signac "RunChromVAR" function was applied to calculate the cell type-specific TF activities and differential activities among cell types were accessed with "FindMarkers" with Bonferroni-adjusted $p < 0.05$.

**Cis-coaccessibility networks generation.** Cis-coaccessibility networks were predicted using the integrated snATAC-seq library in Cicero[19,20]. We isolated the snATAC-seq library into individual cell types and converted them to cell dataset (CDS) objects by using the "make_atac_cds" function. The CDS objects were processed using the "detect_genes" and "estimate_size_factors" functions with default parameters prior to dimensional reduction and conversion to a Cicero CDS object. Cell-type-specific Cicero analysis was conducted using the "run_cicero" function with default parameters.

**snATAC-seq trajectory analysis.** Cicero (v1.3.8) was used to generate trajectories for the snATAC-seq dataset. First, cell dataset (CDS) object was constructed from the peak count matrix in the Seurat object of integrated scATAC-seq data with "make_atac_cds" function. After preprocessing (num_dim = 50), the cells were clustered (cluster_cells) and visualized. We performed cell ordering with the "learn_graph" and "order_cells" in myonuclei. The data was visualized with "plot_cells" function.

**Identify transcription factor hierarchy.** We use Single-Cell regulatory Network Inference and Clustering (SCENIC, version 1.1.2) to predict the potential GRNs regulating normal and denervated muscles[7,28]. We inferred the regulons that consist of upstream TF and its candidate downstream target genes using GENIE3 and analyzed these regulons with RcisTarget. Target genes exhibiting significant motif enrichment of corresponding TF were retained. The genomic regions for TF-motif search were limited to 10 kb around the TSS (mm10). Next, we scored the activity of each regulon in each nucleus with AUCell to identify the active regulons in different cell/nucleus types. Then, we overlapped TFs and regulons that identified as differential enriched in denervated muscle in ChromVar and SCENIC and selected 125 overlapped TFs. Hierarchy height of these TFs was calculated as $h = (O - I)/(O + I)$, where $O$ and $I$ are outgoing and incoming edges of examined TF through SCENIC regulons[54].

**Gene Oncology enrichment.** GO enrichment on the TF target genes was performed by using the "enrichGO" function in the ClusterProfiler. A $p$ value < 0.05 and an adjusted $p$ value < 0.05 was considered statistically significant.

## H3K27ac ChIP-seq

The ChIP seq was prepared using SimpleChIP® Chromatin Immunoprecipitation kit. Briefly, whole gastrocnemius muscles (450 mg) from normal or denervated mice ($n = 3$) were homogenized with dunce grander to isolate myonuclei which were then cross-linked with 1% formaldehyde for 10 min. The myonuclei were treated with 0.5 µl Micrococcal Nuclease for 10 min followed by sonication for 15 min (1/8-inch probe, 30-sec pulses X15 with setting level at 3). After centrifuge, 50 µg cross-linked DNA was used for chromatin immunoprecipitation with anti H3K27ac antibody (Cell Signaling Technology, #8173). The pulled down DNA and input DNA were submitted to LC Sciences LLC (Houston, TX. lcsciences.com) for library preparation and the yield library was sequenced with Illumina platform.

After ChIP-seq sequencing, the obtained raw reads were filtered to remove adapters and contaminations followed by aligning with the reference genome. High-quality mapped reads (MPAQ greater than or equal to 30) were used for subsequent information analysis. Briefly, raw sequencing reads were trimmed to remove adapters and low-quality sequences using Cutadapt and Trimmomatic. The Clean data were quality controlled with FastQC. Clean reads were aligned to the

mouse reference genome (mm10) by bowtie2 (v2.3.5.1). The mapping data were analyzed with the MACS2 (v.2.2.7.1) peak-calling algorithm. High-confidence peak screening between samples using the IDR program. Then, ChIPseeker package was applied for peaks-related annotation. Bigwig files were derived from the output of Deeptools (v3.4.3).

## Bulk RNA-seq

Total RNA from gastrocnemius muscle (100 mg) of normal control mice ($n = 3$) and mice with denervation ($n = 3$) were extracted using QIAzol and precipitated in isopropanol. For Elk4 overexpression bulk RNA-seq, mouse $C_2C_{12}$ cell was harvested with three biological replicates per group. RNA samples that passed quality-control examination were submitted to LC Sciences LLC (Houston, TX. lcsciences.com) for library preparation and paired-end 50 bp sequencing using the standard Illumina mRNA-seq protocol.

Paired-end reads were mapped to the mouse genome using Bowtie2. The abundance was estimated using RSEM v1.3.0 and the differential expression analysis was performed using EdgeR v3.12.1.

## Immunofluorescence staining

Briefly, frozen sections of TA muscles with a thickness of 5 μm were fixed in cold acetone for 5 min, followed by air-drying for 30 min. The tissues were then rinsed twice with Tris-buffered saline containing 0.05% tween-20 (TBS-T). Subsequently, the sections were treated with Dako Protein Block (serum-free, Cat#X0909) for 30 min at room temperature to block any nonspecific binding. After the blocking step, the sections were incubated overnight at 4 °C with primary antibodies diluted in Dako Antibody Diluent (Cat# S3022). The sections were then washed three times for 5 min each with TBS-T, followed by a 30-min incubation at room temperature with Alexa Fluor-conjugated secondary antibodies (either 488 anti-mouse or 546 anti-rabbit, both used at a 1:600 dilution, Invitrogen). After another round of washing with TBS-T three times, the sections were stained with DAPI (Thermo Fisher Scientific, D1306) in TBS-T for 5 min. Fluorescence images were captured using the NIS-Elements system (Nikon)[46]. The antibodies used for both western blotting and immunostaining are listed in the Key Resources Table. To evaluate differences in the cross-sectional areas (CSA) of myofibers, 5 μm sections of TA muscles were stained with an anti-laminin antibody. The CSA of 300 myofibers per TA muscle was measured using Image J software (National Institutes of Health).

## Succinate dehydrogenase (SDH) staining

Transverse sections (5 μm) were cut from the TA muscles at −20 °C and stored at −80 °C until SDH staining was performed. Dried the sections at room temperature for half an hour before incubation in a solution made up of 0.2 M phosphate buffer (pH 7.4), 0.1 M $MgCl_2$, 0.2 M succinic acid (Sigma, St. Louis, MO, USA) and 2.4 mM nitroblue tetrazolium (NBT, Sigma) at 37 °C for 60 min. The sections were then washed in deionized water for 3 min. The sections were then washed in approximate solutions of 30, 60, and 90% acetone to remove unbound NBT. Reduced NBT forms a highly colored formazan dye that is finely granular blue. Digital photographs were taken from each section at 10X magnification under a Nikon Eclipse 80i microscope (Nikon, Melville, NY, USA).

## In vivo enhancer activity and X-gal staining

Candidate enhancers of Smox and Gadd45a were determined identified using Cicero combined with H3K27ac ChIP-seq. We identified two hubs that have co-accessed peaks with the promoter regions of the Smox or Gadd45a genes, respectively. At the same time, these two distal hubs had strong differential H3K27ac signaling between normal and denervated muscles (Fig. 4g, h). We then selected two DNA fragments that cover these two hubs and cloned them into reporter vectors, separately. Detailed sequence and vector information for these two constructs are presented in Supplementary Material 1.

VectorBuilder provides in vivo enhancer reporter vectors and cloning services. The enhancer constructs were electroporated into TA muscles of normal mice or mice with immediately sciatic denervation. After 7 days, the transfected TA muscles were collected and 5 μm cryosections were fixed in cold formalin (4 °C) for 10 min. The slides were washed 3 times for 5 min each time with TBS-T and then incubated with X-gal working solution (5 mM potassium ferricyanide crystalline, 5 mM potassium ferricyanide trihydrate, 2 mM magnesium chloride, 1 mg/ml X-gal in PBS) at 37 °C for 24 h. After washing with TBS-T, the results were reviewed and recorded using Nikon microscope.

## Hematoxylin-eosin staining

Normal and denervated (14 days) mouse gastrocnemius (GAS) muscles were harvested and fixed in 10% buffered formalin for 24 h, embedded in paraffin, and cut into 5 μm thickness with a microtome. After deparaffinization and rehydration, sections were stained with hematoxylin and eosin according to standard protocols. GAS sections were examined and photographed using Nikon microscope.

## Cell culture and treatments

$C_2C_{12}$ myoblasts (ATCC, Manassas, VA) were maintained in DMEM with 10% fetal bovine serum (FBS), penicillin (200 units/ml), and streptomycin (50 μg/ml). Myotubes were induced from myoblast by switching to DMEM plus 2% horse serum (Sigma) for 72 h. siRNA targeting to Elk4 (Dharmacon smartpool siRNA, 0.01 nmol/well in 6-well-dish) was transfected into myotubes with Lipofectamine™ 3000 reagent according to the manufacturer's protocol for 36 h; Scramble siRNA served as the control. All cultured cells were incubated in a humidified 5% $CO_2$ atmosphere at 37 °C.

## Western blot analyses

Total protein of C2C12 cells from 6-well-dish was extracted with 100 μl/well of radio-immunoprecipitation (RIPA) buffer containing protease and phosphatase inhibitor on ice. Skeletal muscle lysates were prepared from ~50 mg muscle by homogenizing in 0.5 ml cold RIPA. After centrifugation ($12,000 \times g$) for 15 min at 4 °C, the yield supernatants were heated with sample loading buffer at 95 °C for 5 min before subjected to SDS-polyacrylamide gel electrophoresis on gradient gels (4–20%). The proteins were electro-transferred to polyvinylidene difluoride membranes. The membranes were blocked with 5% nonfat milk-in TBS-T and incubated overnight with primary antibodies at 4 °C, followed by three times washing and 1 h of incubation with horseradish peroxidase (HRP)-conjugated secondary antibodies at room temperature. The bands were visualized with ChemiDoc MP Imaging System (Bio-Rad Laboratories. Hercules, CA). To ensure equal protein loading, β-actin protein was used as the internal control.

## mRNA preparation and quantitative real-time RT-PCR

Bulk RNA from snap-frozen GAS muscle tissues was extracted with QIAzol lysis reagent (QIAGEN, Germantown, MD) and precipitated with isopropanol. Complementary DNA (cDNA) was synthesized using RT² First Strand Kit (QIAGEN). SYBR Green RT-qPCR was performed with CFX96 Touch Real-Time PCR Detection System (Bio-Rad) according to the manufacturer's instructions. The specificity of RT-PCR was confirmed using melting-curve analysis. The expression levels of the target genes were normalized by β-actin in each sample. The primers for qPCR are listed in Key Resources Table.

## Statistics and reproducibility

Statistical analyses were conducted using the R software or SPSS v26.0 to evaluate the differences and significance between groups in our study. For comparisons between two groups, we utilized a two-tailed unpaired t-test for normal distribution dataset (Fig. 3c) or used Wilcoxon rank-sum test for data that did not adhere to normal distribution (Fig. 6i, j). When comparing more than two groups, we employed

the one-way ANOVA with post hoc Tamhane's multiple comparisons test to assess statistical significance (Fig. 6l, m; Fig. 7a, d–f). Significance was considered if the $p$ value was <0.05.

To elucidate the gene regulatory network in skeletal muscle at single-nucleus precision, we adopted a comprehensive approach that combines single-nucleus and bulk sequencing, along with H3K27ac ChIP-seq. Initially, we focused on pairwise comparisons between normal and denervated conditions in the GAS muscles of male mice. A schematic detailing our integrated methodologies is depicted in Fig. 1a. By integrating the data from these techniques, we developed a detailed atlases of gene regulatory network. Based on the insights gained from this pairwise analysis, we generated several hypotheses about the gene regulatory networks. To validate these new hypotheses, we expanded our experimental design to include multiple group comparisons. Further experiments were conducted both in vitro (using cell cultures) and in vivo (on normal and denervated mice) to observe the effects of "gain" or "loss" function of certain genes. While rigorous data quality checks were imposed, filtering out low-quality nuclei in the single-nucleus analyses (see Method section for details). Male $C57BL6$ mice, age and weight-matched, were randomized into experiment groups. Emphasizing robustness, our single-nucleus experiments were repeated thrice, with in vitro and in vivo tests boasting a minimum of four and five replicates, respectively. No statistical method was used to predetermine sample size and The Investigators were not blinded to allocation during experiments and outcome assessment.

### Reporting summary

Further information on research design is available in the Nature Portfolio Reporting Summary linked to this article.

## Data availability

All primary datasets supporting the findings of this study, including snRNA-seq, bulk RNA-seq, snATAC-seq, and ChIP-seq, have been deposited into the Gene Expression Omnibus (GEO). The datasets and processed data are publicly accessible and can be retrieved using the following accession numbers: GSE183802, GSE217576, GSE217577, For ease of access and immediate reference, we have also provided the essential dataset (minimum dataset) as Supplementary data accompanying our manuscript. We have ensured the transparency and accessibility of our research by incorporating this 'Data availability' section within the Methods part of our manuscript. For the purpose of this study, we did not reference or utilize specific external datasets, nor did we employ data from clinical sources, or any data owned by third parties. Source data are provided with this paper.

## Code availability

Original codes for data analysis were deposited on GitHub at https://github.com/linhch1/Reprogramming-of-Cis-regulatory-networks-during-skeletal-muscle-atrophy.

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

## Acknowledgements

This work supported by National Institute of Health (NIH) grants R01DK037175 and R56AR063686. H.L. is supported by the Applied Basic Research of Guangdong Province (2021B1515230005). H.P. and Y.S. are supported by the National Natural Science Foundation of China (82170762 and 81873613).

## Author contributions

Z.H. conceived and led this study. H.L., H.P., Y.S., and M.S. performed all the single-cell data analysis. H.L., Y.S., and J.W. performed animal and cell culture experiments. H.L. S.S.T. and Z.H. performed histological analysis. Y.W. and Z.S. contributed to the data interpretation. H.P. and Z.S. contributed to experimental design and results analysis. Z.H. wrote the manuscript.

## Competing interests

The authors declare no competing interests.
