## [Peer Review File · Nature Communications]

Reprogramming of *cis*-regulatory networks during skeletal muscle atrophy in male miceREVIEWER COMMENTS

Reviewer #1 (Remarks to the Author):

The study by Lin et al. utilizes matched single-nucleus chromatin accessibility and RNA-sequencing to generate landscapes of the accessible chromatin regions in the gastrocnemius muscle of juvenile (12-week old) male C57BL6 mice under normal and denervated conditions. The manuscript provides a single-cell genomic resource for exploring the regulatory mechanisms underlying skeletal muscle atrophy. Additional studies will need to be performed to determine whether the landscapes developed for denervation-induced atrophy relate to other forms of muscle atrophy such as disuse, cachexia and sarcopenia.

The identification of ELK1 as an atrophy-related transcription factor that promotes muscle atrophy through the regulation of TGF- β 1 is novel and significant. The genomic data is strengthened by the in vivo experiments and the data presented in Figure 6. A limitation of the in vivo electroporation data are that they were performed only in the TA muscle.

Specific comments:

Title: The title needs to specify that these data came from male mice.

Abstract: The abstract needs to identify that these data were collected in juvenile (12 weeks) male C57BL6 mice. Also, the abstract states that you "examined the regulatory circuits that underpin the transition between oxidative and glycolytic myofibers". However, what was examined was the transition between myofibers expressing different myosin heavy chain isoforms. Muscle fibers differ in their metabolic and contractile properties. While it is true that in normal adult muscle there is a relationship between myosin heavy chain expression and oxidative (mitochondria) properties such that type I and IIa fibers are highly oxidative and type IIb fibers have the lowest oxidative properties; this relationship can change in response to exercise training or pathological conditions that induce muscle atrophy. It is incorrect to use glycolytic and oxidative to infer myosin heavy chain expression.

Line 80: "normal mouse skeletal muscle"-----It should be clearly stated that the atlas that was generated came from the gastrocnemius muscle of 12-week old male C57BL6 mice .

Line 152: This analysis is examining the transition between myosin heavy chain expression in myofibers, not the change in oxidative capacity. Type IIb fibers can increase their oxidative capacity in response to endurance training without changing MHC expression. Fibers can also decrease their oxidative capacity without changing their MHC expression.

Line 188-200: This analysis is not examining fiber type switching. This analysis is from fibers that are under a static condition. It is examining the relationships that have been established as a function of development and maturation to define the properties of fibers expressing different myosin heavy chain types. It is generally true that in normal adult muscle, type IIa fibers are more oxidative than IIb fibers. However, one should not predict metabolic properties from MHC properties.

Line 346: Denervation does not stimulate a glycolytic to oxidative myofiber transition. The oxidative capacity of the muscle decreases following denervation. Again, do not conflate MHC isoform expression with oxidative capacity.

Reviewer #2 (Remarks to the Author):

Authors sought to determine transcriptomic, chromatin accessibility, and epigenetic changes at single-nucleus resolution to generate a comprehensive atlas of interaction between the muscle-resident cell populations during muscle neurogenic atrophy.

Authors used an established model of 1-/2-week-long denervation of the gastrocnemius muscle in the mouse.

Authors used separate single-nucleus RNA-Seq and ATAC-Seq, bulk RNA-Seq, ChIP-Seq for H3K27ac, and extensive bio-informatic analysis to identify cell-type specific regulatory networks acting in cis, and gene expression/signaling regulatory circuits underlying the interactions between myofibers and muscle-resident cells, controlling myofiber identity, and regulating the reprogramming of muscle metabolic features post-denervation.

Authors identified novel roles of specific transcription factors in regulating muscle neurogenic atrophy.

Major discoveries/results:

- 1) The generation of an atlas of regulatory networks by combining results of gene expression and chromatin accessibility of promoters and enhancers in eleven different types of muscle-resident cells after muscle denervation.
- 2) The identification of specific cis-regulatory elements (CREs) regulating differential accessible chromatin regions (ACRs) nearby the transcription start sites (TSSs) in eight muscle-resident cells. Authors show that more than 60% of ACRs are situated in proximal promoter regions or in distal enhancers.
- 3) The finding of more than 50,000 cis-regulatory links underlying co-accessibility between ACRs (potential enhancers).
- 4) The reduction of ACRs distribution in the promoter regions of myonuclei upon denervation coupled by a parallel ACRs increment in their distal and intragenic regions.
- 5) Myonuclei of the denervated muscle showing the highest level of differentially accessible regions (DARs) in enhancer regions linked to the expression of muscle atrophy-related genes, and the parallel reduction of distal DARs regulating genes for sarcomeric proteins.
- 6) The identification of atrophy-responsive TFs upon muscle denervation affecting several muscle-resident cell populations, and the main affected regulons in myonuclei of denervated Gas.
- 7) The analysis of TFs landscape upon Gas denervation complemented with the H3K27ac genomic distribution to identify the epigenetic modulation of new candidate enhancers in the different muscle-resident cell populations.
- 8) Generation of a myofiber-specific TFs hierarchy upon neurogenic atrophy induction. This may become a reference model useful to follow and predict the evolution of epigenetic and transcriptomic changes in different models of muscle atrophy.
- 9) The identification of Elk4 as a major pro-atrophic/catabolic driver in the denervated muscle.

Overall, the study is well designed and conducted, using the appropriate methods of analysis.

I have minor comments and the request of some editing as follows:

- 1) Please better specify all over text and figure legends when the analysis was carried out on muscles TAs and Gas harvested after 7 and 14 days from denervation.
- 2) Throughout the figures I would state in the legends the magnification for all the muscle section IF images.
- 3) Lines 125-133. Considering the evaluation of some pioneer transcription factors (PTFs) potentially involved in CREs activation would be interesting. In addition, a description of what is Tn5, also in Fig. 1g, would be helpful.
- 4) Lines 152-154. Considering changing the wording of this sentence or just list nerve activity, exercise, or hormonal influences instead of "nerve activity or by exercise or hormonal influences".
- 5) Line 186. Please describe in text and Fig. 2h legend which muscle is depicted, TA? Gas? Normal or denervated?
Maybe add a co-staining for MyHC-I or MyHC-II for fiber typing.
- 6) Line 209. Authors stated that denervated fibers show lower level of heterochromatin; based on which type of assay? Are these two images at the same magnification? Maybe an H&E would make it easier to see the atrophy. Also, the evidence of nucleoli by EM is not fully convincing.
- 7) Line 214. In text Authors state ACRs, while in Fig. 3e legend differentially accessible regions

(DARs). Please clarify.

8) Line 219. Authors write denervated muscle, while the percentage shown in Fig. 3e is referring to myonuclei. Muscle might be perceived as a whole in this manner, including all the 11 cell populations, so please clarify in the text. In addition, always in Fig. 3e, because many other types of muscle-resident cells show little changes between normal vs. denervated, or changes in the opposite direction when compared to what is shown by myonuclei, it would be great to add a comment on that.

9) Line 244. Skeletal structural components; please change to skeletal muscle structural components.

10) Line 264. DARs with reduced expression. Is it better to say DARs with reduced expression of their linked genes? The same in Line 267.

11) Line 272. Extended data fig.4; please specify Fig. 4b-g.

12) Line 301. It is interesting the role of Ar with its strong downregulation upon denervation, as well as that of its target genes. Authors used male mice in this study; please add an additional comment on that in the discussion section.

13) Extended Fig. 6g legend: oxidative capacity IN normal and...

14) Line 324. Author shows X-gal staining in mouse TA but does not comment on this in the text while looking at Smox and Gadd45a expression following denervation. Are we seeing increase senescence in these fibers with denervation with Gadd45a?

15) Lines 356-8. Authors wrote: "increasing motif activity at beginning of denervation followed by a decline as denervation progresses". Even if Authors are talking also about a pseudotime, this statement indicates that Authors conducted the analysis at different time points after denervation as stated in Methods, in Fig. 6c "7 days and 14 days", and in Line 402 "...as denervation progressed. Please reformulate these statements more clearly and consistently.

16) Fig. 5g LogFC denervation vs. normal; please edit.

17) Fig. 6I text ELH4; please change to ELK4.

18) How can Authors comment the partial evidence in Fig. 6k that Elk4 knockdown does not increase C2C12 myotube size? The same question on Fig. 7p in vivo. I do agree that Authors did not find myostatin activation upon muscle denervation, but this evidence requires an additional comment.

Since it seems that Elk4 does not bind autonomously to DNA, how Authors explain Elk4 action on chromatin and on its target genes? It might be mediated by serum response factor (SRF) TF?

19) Could ionomycin induce pro-atrophic pathways independent of Elk4 action?

20) In Extended Fig. 7d it is not possible to appreciate myofibers size. Please reformulate that statement in Fig. 7d legend.

21) It is possible to add the total Smad3 in the Western blotting of Fig. 6l and Fig. 6o?

22) What does represent the red color in Fig. 6p? Laminin?

23) Lines 571-72. Please edit sentence "Mice underwent surgical of denervation was described previously".

24) Line 600. About 10,000 cells? Probably 10,000 nuclei.

25) Lines 619-20. Please better describe how data have been processed and analyzed.

26) Line 729. Please add the processing for IFs for Gas, since something is missing somewhere, see Fig. 2h and Fig. 3a.

27) Lines 749-752. Please better describe the staining. Prepare? And remove? Leave?

28) Line 770. Please add AAV1-GFP as per Fig. 6k.

29) Lines 773-777. When were the cells transfected with siRNA? Was it following 96 hrs of differentiation? How long were the siRNAs transfected before processing of myotubes?

30) Line 792. How much tissue from the Gas was used for RNA extraction?

Reviewer #3 (Remarks to the Author):

In this study, the authors profiled landscapes of the accessible chromatin regions in skeletal muscles of normal and denervated mice, identified cell-type-specific cis-regulatory networks, illustrated the reprogramming of cis-regulatory networks in response to denervation, and revealed the interplay of key promoters/enhancers and target genes. This is a meaningful study, which provides a theoretical basis for studying physiological and pathological metabolism in skeletal

muscle, as well as facilitates further hypothesis- or data-driven research.

The study is well designed and optimally organized. Three biological replicates make the data and of single-nucleus chromatin accessibility and RNA-sequencing fair and solid. The research on the regulatory function of ELK4 in denervated muscles further confirms the reliability and value of the foregoing analysis.

However, there are some issues that should be addressed before considering publication in Nature communications.

Major comments:

1. Line 117: Figure 1e does not confirm the specific expression of these genes, but only that they are expressed in normal Gas muscle.
2. Figure 1g: Authors should provide UMAP plots showing the distribution of all cell types. Otherwise, cell-type-specific activities of TFs motif cannot be distinguished.
3. Line 156: Is the data used to analyze the myofiber type transition only from normal muscles? The authors should explain the significance of studying the myofiber type transition only in normal adult mice.
4. Authors should add scale bars to all histological and fluorescent-stained photographs.
5. Figure 2h: The immunofluorescence co-staining assays of these three differentially accessible genes with myofiber type markers (MYH 4,7, and1) should be performed to illustrate their dynamic expression changes in the process of myofiber type transition.
6. Figure 3h: Does purple lines indicate co-accessibility link score in normal myonuclei or in denervated myonuclei? How does the authors compare the difference in distance regulation between normal myonuclei and denervated myonuclei?
7. Lines 358-362: Does the beginning of the pseudotime trajectory represent the "beginning of denervation"? If the answer is "yes", it can be seen from Figure 5f that the motif activity of NR3C1 does not increase first and then decrease along with denervation, nor does the motif activity of MYOG increase persistently with denervation. If the answer is "no", what point in the pseudotime trajectory is the "beginning of denervation"?
8. Line 376: How does the gene regulatory network with TF hierarchy exhibit a divergent pattern over time?
9. In Extended Data Fig. 6, SDH staining of denervated group appears to be selective.

Minor comments:

1. Line 301: "Figure 5c" should be "Figure 4c".

REVIEWER COMMENTS

Reviewer #1 (Remarks to the Author):

The study by Lin et al. utilizes matched single-nucleus chromatin accessibility and RNA-sequencing to generate landscapes of the accessible chromatin regions in the gastrocnemius muscle of juvenile (12-week-old) male C57BL6 mice under normal and denervated conditions. The manuscript provides a single-cell genomic resource for exploring the regulatory mechanisms underlying skeletal muscle atrophy. Additional studies will need to be performed to determine whether the landscapes developed for denervation-induced atrophy relate to other forms of muscle atrophy such as disuse, cachexia and sarcopenia.

Response: We greatly appreciate your insightful suggestions. We concur that different forms of muscle atrophy likely share certain commonalities while also exhibiting unique molecular mechanisms at the cellular level. Prior research, including the studies by Nicole Almanzar et al. (Nature, 2020) and Deirdre D Scripture-Adams et al. (Commun Biol, 2022), has characterized single-cell RNA-seq gene expression profiles of muscle atrophy caused by aging and Duchenne Muscular Dystrophy (DMD). Our study indeed adds to this growing body of literature by providing a single-cell genomic resource specific to denervation-induced muscle atrophy. As you pointed out, understanding the potential common and unique regulatory landscapes across different forms of muscle atrophy such as disuse, cachexia, sarcopenia, and atrophy associated with chronic conditions like cancer and chronic kidney disease, would indeed be valuable. We appreciate your insights regarding the need for additional studies on this topic. We fully agree and are committed to pursuing further research in this area. The methodologies and results from our current study will significantly guide these future investigations.

The identification of ELK1 as an atrophy-related transcription factor that promotes muscle atrophy through the regulation of TGF- β 1 is novel and significant. The genomic data is strengthened by the in vivo experiments and the data presented in Figure 6. A limitation of the in vivo electroporation data are that they were performed only in the TA muscle.

Response: We greatly value your insight on the limitation of our study involving the in vivo electroporation data limited to the TA muscle, especially, it did not provide the insights of how ELK4 influence whole body metabolism and muscle hemostasis. We currently do not possess ELK4 knockout mice. However, recognizing the importance of this, we plan to utilize the LoxP-Cre technique to create ELK4 knockout in skeletal muscles in transgenic mice for more comprehensive studies. We intend to investigate the systemic impact of ELK4 on protein and energy metabolism. We are optimistic that these future investigations will further our understanding of the multifaceted roles of ELK4 in muscle atrophy and metabolism.

Specific comments:

Title: The title needs to specify that these data came from male mice.

Response: We appreciate your suggestion to specify the gender of the mice used in our study. In line with your advice, we have revised the title to: "Reprogramming of cis-regulatory networks during skeletal muscle atrophy in male mice".

Abstract: The abstract needs to identify that these data were collected in juvenile (12 weeks) male C57BL6 mice. Also, the abstract states that you "examined the regulatory circuits that underpin the transition between oxidative and glycolytic myofibers". However, what was examined was the transition between myofibers expressing different myosin heavy chain isoforms. Muscle fibers differ in their metabolic and contractile

properties. While it is true that in normal adult muscle there is a relationship between myosin heavy chain expression and oxidative (mitochondria) properties such that type I and IIa fibers are highly oxidative and type IIb fibers have the lowest oxidative properties; this relationship can change in response to exercise training or pathological conditions that induce muscle atrophy. It is incorrect to use glycolytic and oxidative to infer myosin heavy chain expression.

Response: We appreciate the reviewer's guidance. We have revised the phrase "examined the regulatory circuits that underpin the transition between oxidative and glycolytic myofibers" to "examined the regulatory circuits that underpin the transition between different myonuclear types". Additionally, we have included information on the gender, age, and strain of the mice used in our study in the abstract. Please refer to the updated abstract in our revised manuscript.

Line 80: "normal mouse skeletal muscle" -----It should be clearly stated that the atlas that was generated came from the gastrocnemius muscle of 12-week old male C57BL6 mice.

Response: We appreciate your suggestion for specificity. We have now revised the text on line 80 from "normal mouse skeletal muscles and those with neurogenic atrophy" to "normal gastrocnemius muscles and those undergoing neurogenic atrophy in 12-week-old male C57BL6 mice". Please see red text in page 3 line 77.

Line 152: This analysis is examining the transition between myosin heavy chain expression in myofibers, not the change in oxidative capacity. Type IIb fibers can increase their oxidative capacity in response to endurance training without changing MHC expression. Fibers can also decrease their oxidative capacity without changing their MHC expression.

Response: We thank you for this important clarification. We have updated the text on line 148 from "Evidence has shown switching of oxidative (type IIa/IIx) myofiber to glycolytic (type IIb) myofiber" to "Evidence has demonstrated the transition of myosin heavy chain expression between slow (type IIa/IIx) myofibers and fast (type IIb) myofibers." Please refer to the revised manuscript for this update.

Line 188-200: This analysis is not examining fiber type switching. This analysis is from fibers that are under a static condition. It is examining the relationships that have been established as a function of development and maturation to define the properties of fibers expressing different myosin heavy chain types. It is generally true that in normal adult muscle, type IIa fibers are more oxidative than IIb fibers. However, one should not predict metabolic properties from MHC properties.

Response: We agree with your observation that the metabolic characteristics of skeletal muscle cannot be directly equated to MHC properties. Previous studies have indeed shown that myofiber type can be influenced by factors such as nerve activity, exercise, and hormonal changes. However, the cis-regulatory networks governing the configuration of MHCs and metabolic properties under normal conditions are still not fully understood. To address this, we performed a trajectory analysis and observed an ordered transition from type IIa to type IIx and from type IIx to type IIb within the myonuclei of normal muscles. This led us to hypothesize a dynamic equilibrium existing between different myonuclear-types under normal conditions, and this equilibrium is necessary for maintaining myofiber-type configuration and muscle function. Similar principles apply to the regulation of energy metabolic enzymes. To avoid confusion, we've reframed our description in the manuscript, opting for the term "myonuclear transition" rather than "myofiber type switching". We have made the appropriate revisions to our manuscript to reflect these clarifications.

Thank you again for your astute observations and suggestions, which have helped us improve the clarity and accuracy of our work.

Line 346: Denervation does not stimulate a glycolytic to oxidative myofiber transition. The oxidative capacity of the muscle decreases following denervation. Again, do not conflate MHC isoform expression with oxidative capacity.

Response: We appreciate your important clarification regarding MHC isoform expression and oxidative capacity. In response, we have revised our manuscript, replacing "oxidative myofiber" with "type IIa myofiber" on Page 13, Line 345. Our observation of a shift in MHC isoform expression, specifically from Myh4 to Myh2 during denervation-induced muscle atrophy, aligns with previous studies. For instance, Matthieu Dos Santos et al. demonstrated the endogenous spatio-temporal expression of adult Myosin genes in normal skeletal muscle using a transgenic model with a super-enhancer at the locus containing Myosin genes, including Myh2 and Myh4 (A fast Myosin super enhancer dictates muscle fiber phenotype through competitive interactions with Myosin genes. Nat Commun. 2022). Our claim about changes in Myh expression is corroborated by our bulk RNA-seq, snRNA-seq, and snATAC-seq data. We propose that these changes may contribute to the functional disorder of myofibers seen during muscle atrophy. We thank you for raising this point and enabling us to improve the clarity and precision of our manuscript.

Reviewer #2 (Remarks to the Author):

Authors sought to determine transcriptomic, chromatin accessibility, and epigenetic changes at single-nucleus resolution to generate a comprehensive atlas of interaction between the muscle-resident cell populations during muscle neurogenic atrophy.

Authors used an established model of 1-/2-week-long denervation of the gastrocnemius muscle in the mouse.

Authors used separate single-nucleus RNA-Seq and ATAC-Seq, bulk RNA-Seq, ChIP-Seq for H3K27ac, and extensive bio-informatic analysis to identify cell-type specific regulatory networks acting in cis, and gene expression/signaling regulatory circuits underlying the interactions between myofibers and muscle-resident cells, controlling myofiber identity, and regulating the reprogramming of muscle metabolic features post-denervation.

Authors identified novel roles of specific transcription factors in regulating muscle neurogenic atrophy.

Major discoveries/results:

- 1) The generation of an atlas of regulatory networks by combining results of gene expression and chromatin accessibility of promoters and enhancers in eleven different types of muscle-resident cells after muscle denervation.
- 2) The identification of specific cis-regulatory elements (CREs) regulating differential accessible chromatin regions (ACRs) nearby the transcription start sites (TSSs) in eight muscle-resident cells. Authors show that more than 60% of ACRs are situated in proximal promoter regions or in distal enhancers.
- 3) The finding of more than 50,000 cis-regulatory links underlying co-accessibility between ACRs (potential enhancers).
- 4) The reduction of ACRs distribution in the promoter regions of myonuclei upon denervation coupled by a parallel ACRs increment in their distal and intragenic regions.
- 5) Myonuclei of the denervated muscle showing the highest level of differentially accessible regions (DARs) in enhancer regions linked to the expression of muscle atrophy-related genes, and the parallel reduction of distal DARs regulating genes for sarcomeric proteins.
- 6) The identification of atrophy-responsive TFs upon muscle denervation affecting several muscle-resident cell populations, and the main affected regulons in myonuclei of denervated Gas.
- 7) The analysis of TFs landscape upon Gas denervation complemented with the H3K27ac genomic distribution to identify the epigenetic modulation of new candidate enhancers in the different muscle-resident

cell populations.

8) Generation of a myofiber-specific TFs hierarchy upon neurogenic atrophy induction. This may become a reference model useful to follow and predict the evolution of epigenetic and transcriptomic changes in different models of muscle atrophy.

9) The identification of Elk4 as a major pro-atrophic/catabolic driver in the denervated muscle.

Overall, the study is well designed and conducted, using the appropriate methods of analysis.

Response: We sincerely appreciate the reviewer's positive comments and thoughtful insights. They will undoubtedly contribute to enhancing the quality of our work. We will address all the concerns raised with our greatest attention and effort.

I have minor comments and the request of some editing as follows:

1) Please better specify all over text and figure legends when the analysis was carried out on muscles TAs and Gas harvested after 7 and 14 days from denervation.

Response: We have thoroughly reviewed and revised the text and figure legends to clearly specify when the analysis was carried out on TA and Gas muscles harvested at 7- and 14-days post-denervation. Please refer to the updated manuscript for these changes.

2) Throughout the figures I would state in the legends the magnification for all the muscle section IF images.

Response: We have now included scale bars and stated the magnification for all immunofluorescence images of muscle sections in the figure legends. Please refer to the updated figures and their corresponding legends.

3) Lines 125-133. Considering the evaluation of some pioneer transcription factors (PTFs) potentially involved in CREs activation would be interesting. In addition, a description of what is Tn5, also in Fig. 1g, would be helpful.

Response: Thank you for the suggestion. In Fig.1g, we presented Myod1 and Myf5, which are two important pioneer transcription factors (PTFs) known in skeletal muscle. We have now also added other PTFs such as Six1 and Klf4 to Fig.S2f. Furthermore, Tn5 transposase is the DNA-modifying enzyme to generate snATAC-seq. Tn5 enrichment analysis involves assessing the frequency or density of Tn5 transposase insertions at different genomic regions, therefore can be used to inference the TF binding site. We've expanded the description of Tn5 enrichment analysis in the manuscript as: Utilizing Tn5 enrichment analysis, a method designed to pinpoint ACRs targeted by TF binding, we observed a "summit depletion in peaks" across all selected TF motifs. This indicates that the physical binding of TF to the motif impedes the integration of Tn5 transposase (Page 5, line 126-130).

4) Lines 152-154. Considering changing the wording of this sentence or just list nerve activity, exercise, or hormonal influences instead of "nerve activity or by exercise or hormonal influences".

Response: We appreciate your advice and have revised the sentence to streamline the expression to "nerve activity, exercise, or hormonal influences" (page 6, line 150).

5) Line 186. Please describe in text and Fig. 2h legend which muscle is depicted, TA? Gas? Normal or denervated?

Maybe add a co-staining for MyHC-I or MyHC-II for fiber typing.

Response: We carried out the immunostaining in Gas muscles, as suggested. Detailed descriptions of the samples have been added to both the main text and the figure legend. We have also incorporated a co-staining for Myh2 (Type IIa) or Myh4 (Type IIb) for fiber typing. Please see revised Fig.2h.

6) Line 209. Authors stated that denervated fibers show lower level of heterochromatin; based on which type of assay? Are these two images at the same magnification? Maybe an H&E would make it easier to see the atrophy. Also, the evidence of nucleoli by EM is not fully convincing.

Response: Like previous electron microscopy studies, which revealed that heterochromatin preferentially localizes to the nuclear periphery and around the nucleolus (J Padeken, P Heun - Current opinion in cell biology, 2014), our assertion that denervated fibers exhibit lower levels of heterochromatin was based on electron microscopy. As suggested, we've performed H&E staining to illustrate muscle atrophy and replaced the EM image with one of higher resolution to offer more convincing evidence. The H&E staining clearly presents changes in the nuclei of male mice muscles during muscle atrophy (Fig.S4a).

7) Line 214. In text Authors state ACRs, while in Fig. 3e legend differentially accessible regions (DARs). Please clarify.

Response: We apologize for the lack of clarity. In Fig. 3e, the accurate term should be differentially accessible regions (DARs) across all cell types. We have corrected and clarified this in the revised manuscript (Page 8, line 218).

8) Line 219. Authors write denervated muscle, while the percentage shown in Fig. 3e is referring to myonuclei. Muscle might be perceived as a whole in this manner, including all the 11 cell populations, so please clarify in the text. In addition, always in Fig. 3e, because many other types of muscle-resident cells show little changes between normal vs. denervated, or changes in the opposite direction when compared to what is shown by myonuclei, it would be great to add a comment on that.

Response: We apologize for the imprecise expression. We have revised the term "muscle" to "myonuclei". Furthermore, we have added a discussion on the changes in the location of DARs in other types of muscle-resident cell nuclei. Please refer to the updated manuscript (Page 8, line 216).

9) Line 244. Skeletal structural components; please change to skeletal muscle structural components.

Response: We have revised the term "skeletal structural components" to "skeletal muscle structural components" in our manuscript (Page 9, line 243).

10) Line 264. DARs with reduced expression. Is it better to say DARs with reduced expression of their linked genes? The same in Line 267.

Response: We appreciate the feedback and have revised the phrase "DARs with reduced expression" to "DARs with reduced expression of their linked genes".

11) Line 272. Extended data fig.4; please specify Fig. 4b-g.

Response: We have updated the reference from "fig.4" to "fig.4b-g".

12) Line 301. It is interesting the role of Ar with its strong downregulation upon denervation, as well as that of its target genes. Authors used male mice in this study; please add an additional comment on that in the discussion section.

Response: As suggested, we added a comment on page 11, line 299.

13) Extended Fig. 6g legend: oxidative capacity IN normal and...

Response: We have corrected the legend to read as “oxidative capacity in normal and denervated (14 days) TA muscle”.

14) Line 324. Author shows X-gal staining in mouse TA but does not comment on this in the text while looking at Smox and Gadd45a expression following denervation. Are we seeing increase senescence in these fibers with denervation with Gadd45a?

Response: The X-gal result presented is an *in vivo* enhancer assay rather than an evaluation of senescence. In this assay, the reporter LacZ gene is placed under the control of regulatory elements corresponding to the enhancers of the genes of interest, such as Gadd45a and Smox. The expression of LacZ is indicated by a dark green stain, providing a robust visual readout of enhancer activity in the TA muscle. The results of *in vivo* enhancer assay supported our enhancer inference using Cicero analysis. We have revised these sentences for clarity.

15) Lines 356-8. Authors wrote: “increasing motif activity at beginning of denervation followed by a decline as denervation progresses”. Even if Authors are talking also about a pseudotime, this statement indicates that Authors conducted the analysis at different time points after denervation as stated in Methods, in Fig. 6c “7 days and 14 days”, and in Line 402 “...as denervation progressed. Please reformulate these statements more clearly and consistently.

Response: We carried out the pseudotime analysis at two time points: normal and denervation at 14 days. In Fig.6c, we evaluated muscle samples at denervation 7 days and 14 days. To avoid any potential confusion, we have decided to remove the 7-day denervation data in Fig.6c from the manuscript.

16) Fig. 5g LogFC denverion vs. nnormal; please edit.

Response: We apologize for the typographical error. We have corrected the legend to read "LogFC denervation vs. normal" in Fig.5g.

17) Fig. 6l text ELH4; please change to ELK4.

Response: We regret the error and have now corrected the term to "ELK4" in our manuscript.

18) How can Authors comment the partial evidence in Fig. 6k that Elk4 knockdown does not increase C2C12 myotube size? The same question on Fig. 7p *in vivo*. I do agree that Authors did not find myostatin activation upon muscle denervation, but this evidence requires an additional comment. Since it seems that Elk4 does not bind autonomously to DNA, how Authors explain Elk4 action on chromatin and on its target genes? It might be mediated by serum response factor (SRF) TF?

Response: In Fig.6k, we did not observe a significant increase in C2C12 myotube size after applying Elk4-siRNA. We have supplemented this data with quantified images and values in Fig.S7e. In the *in vivo*

experiments depicted in Fig.6p, we have also included a quantitative schematic diagram in Fig.7d. Recent data indicate that the ELK family contains DNA binding domains (Elk4:MA0076.2, Elk3: MA0759.1, Elk1: MA0028.1). There is research suggesting cooperative regulation between Elk4 and MLK1/2 on SRF in macrophages (Xie, L. BMC Genomics. 2014). However, due to the unavailability of a suitable antibody for Elk4 ChIP, we were unable to directly confirm this hypothesis. We acknowledge that the exploration of ELK4's role in muscle atrophy is just beginning and certainly warrants further investigation. We hope to elucidate many of the mysteries surrounding ELK4 in future research.

19) Could ionomycin induce pro-atrophic pathways independent of Elk4 action?

Response: Our in vitro experimental evidence indicates that the effect of ionomycin is reduced upon knocking down the expression of Elk4 in C2C12 cells. It seems that ionomycin primarily relies on Elk4 action.

20) In Extended Fig. 7d it is not possible to appreciate myofibers size. Please reformulate that statement in Fig. 7d legend.

Response: We have removed the mention of “myofiber size” in the legend of Extended Fig. 7d.

21) It is possible to add the total Smad3 in the Western blotting of Fig. 6l and Fig. 6o?

Response: Yes. We have included the total Smad3 in the Western blotting results in Fig.6l and Fig.6o. Please refer to Figure 6.

22) What does represent the red color in Fig. 6p? Laminin?

Response: We apologize for the lack of clarity. The red signal in Fig.6p represents Laminin. We have clarified this in the revised Fig .7b and its legend.

23) Lines 571-72. Please edit sentence “Mice underwent surgical of denervation was described previously”.

Response: We have revised the sentence to read, "The procedures for denervation surgery were performed as previously described”.

24) Line 600. About 10,000 cells? Probably 10,000 nuclei.

Response: We have corrected the term "cells" to "nuclei".

25) Lines 619-20. Please better describe how data have been processed and analyzed.

Response: We have provided a more detailed explanation of the data processing and analysis methods in the revised manuscript.

26) Line 729. Please add the processing for IFs for Gas, since something is missing somewhere, see Fig. 2h and Fig. 3a.

Response: We've supplemented the IF co-staining procedure in the method section to address the reviewer's inquiry regarding Fig.2h.

27) Lines 749-752. Please better describe the staining. Prepare? And remove? Leave?

Response: We've revised the SDH staining methodology to enhance clarity. The updated paragraph is as follows: Transverse sections (5 μ m) were cut from the TA muscles at -20°C and stored at -80°C until SDH staining was carried out. Sections were dried at room temperature for 30 minutes before incubation in a solution containing 0.2 M phosphate buffer (pH 7.4), 0.1 M MgCl₂, 0.2 M succinic acid (Sigma, St. Louis, MO, USA), and 2.4 mM nitroblue tetrazolium (NBT, Sigma) at 37°C for 60 minutes. Following this, the sections were rinsed in deionized water for three minutes. Then, they were washed in successive concentrations of acetone (30, 60, and 90%) to remove unbound NBT. Reduced NBT forms a finely granular blue formazan dye. Digital photographs were taken at 10X magnification under a Nikon Eclipse 80i microscope (Nikon, Melville, NY, USA).

28) Line 770. Please add AAV1-GFP as per Fig. 6k.

Response: We've incorporated AAV1-GFP in Fig.6k.

29) Lines 773-777. When were the cells transfected with siRNA? Was it following 96 hrs of differentiation? How long were the siRNAs transfected before processing of myotubes?

Response: The C2C12 cells underwent siRNA transfection 72 hrs post-differentiation. The C2C12 myotubes were processed 36 hrs post-siRNA transfection. We've elaborated on the cell processing methodology in the manuscript. Please refer to the updated version.

30) Line 792. How much tissue from the Gas was used for RNA extraction?

Response: We utilized 100mg of Gas muscle tissue for RNA extraction. This detail has been included in the methods section. Please refer to the revised manuscript.

Reviewer #3 (Remarks to the Author):

In this study, the authors profiled landscapes of the accessible chromatin regions in skeletal muscles of normal and denervated mice, identified cell-type-specific cis-regulatory networks, illustrated the reprogramming of cis-regulatory networks in response to denervation, and revealed the interplay of key promoters/enhancers and target genes. This is a meaningful study, which provides a theoretical basis for studying physiological and pathological metabolism in skeletal muscle, as well as facilitates further hypothesis- or data-driven research. The study is well designed and optimally organized. Three biological replicates make the data and of single-nucleus chromatin accessibility and RNA-sequencing fair and solid. The research on the regulatory function of ELK4 in denervated muscles further confirms the reliability and value of the foregoing analysis. However, there are some issues that should be addressed before considering publication in Nature communications.

Response: We are grateful for the reviewer's commendation. We will make every effort to address reviewer's all thoughtful points and concerns adequately.

Major comments:

1. Line 117: Figure 1e does not confirm the specific expression of these genes, but only that they are expressed in normal Gas muscle.

Response: We appreciate the reviewer's comment. We have revised the results accordingly by adjusting the description or adding new images with corresponding markers. For Trim63 and Prkag3, which are expressed in all types of myofibers, either in the cytoplasm or under the cell membrane, we have kept the original image as it adequately represents this point. For Chodl, it is primarily expressed in MuSCs (or satellite cells). In the updated image, its expression is colocalized with Pax7. For Scara5, we found its signal located uniquely within interstitial space, corroborating the result that its expression is mainly in FAPs.

2. Figure 1g: Authors should provide UMAP plots showing the distribution of all cell types. Otherwise, cell-type-specific activities of TFs motif cannot be distinguished.

Response: We appreciate the reviewer's suggestion. We have labeled the cell types that exhibit the most motif activities, and for comprehensiveness, we have provided detailed UMAP plots in the supplementary data (FigS2.g) to illustrate all cell types.

3. Line 156: Is the data used to analyze the myofiber type transition only from normal muscles? The authors should explain the significance of studying the myofiber type transition only in normal adult mice.

Response: Indeed, our data includes analysis of "myonuclear transition", which we believe may regulate myofiber type configuration. Our trajectory analysis illustrates a continuous arrangement of myonuclei from type IIa to IIb along the pseudotime axis, demonstrating overlap and suggesting a possible continuum in normal skeletal muscle. This not only provides transcriptional evidence for the existence of hybrid myofibers but also points to potential myonuclear transitions between type IIa and IIx or IIx and IIb. We have revised our Results 2 to clarify that this observed "myonuclear transition" is distinct from myofiber switching, but potentially forms its foundation. We greatly appreciate your insightful question that allowed us to refine and improve the precision of our study.

4. Authors should add scale bars to all histological and fluorescent-stained photographs.

Response: We have added scale bars to all histological and fluorescent-stained photographs in the revised manuscript as per the reviewer's recommendation.

5. Figure 2h: The immunofluorescence co-staining assays of these three differentially accessible genes with myofiber type markers (MYH 4,7, and 1) should be performed to illustrate their dynamic expression changes in the process of myofiber type transition.

Response: Acknowledging the reviewer's suggestion, we performed co-staining with myofiber markers (Myh2 and Myh4) to illustrate the myofiber type. The results are included in revised Figure 2h.

6. Figure 3h: Does purple lines indicate co-accessibility link score in normal myonuclei or in denervated myonuclei? How does the authors compare the difference in distance regulation between normal myonuclei and denervated myonuclei?

Response: The purple lines signify the co-accessibility link score in a single nuclei-type, encompassing both normal and denervated conditions. To provide a clearer understanding, we adjusted the cicero process to identify links in the cell type of normal or denervation separately and presented the results in the revised manuscript and Figure 3h and 3k.

7. Lines 358-362: Does the beginning of the pseudotime trajectory represent the "beginning of denervation"? If the answer is "yes", it can be seen from Figure 5f that the motif activity of NR3C1 does not increase first

and then decrease along with denervation, nor does the motif activity of MYOG increase persistently with denervation. If the answer is “no”, what point in the pseudotime trajectory is the "beginning of denervation"?

Response: The beginning of the pseudotime trajectory does not represent the "beginning of denervation," but rather the "start of normal myonuclei." This encompasses myonuclei from both normal muscle and 14-days denervated muscle. It's important to note that even within the denervated muscle, there exist fractions of myonuclei that retain normal transcriptional status or are “still on the way to denervation,” illustrating the ongoing process. The figure below (please refer to the revised manuscript) shows the change in cell numbers (density) along the pseudotime axis, which includes all nuclei, both normal and denervated. Each nucleus is assigned a unique pseudotime number to depict its progression along the transition continuum. We hope this clarifies your concerns, and we appreciate your keen attention to detail.

Density plot for calculating the cell number along the pseudotime.

8. Line 376: How does the gene regulatory network with TF hierarchy exhibit a divergent pattern over time?

Response: We appreciate your inquiry. In our study, we categorized TFs based on their hierarchy scores, derived from SCENIC analysis, indicating the influence each TF exerts within the network. After denervation, we observed an extensive DNA opening and reactivation of numerous TFs, many of which belong to the regulon of high-tier TFs, culminating in a divergent network pattern. Please note that this pattern reflects a snapshot at 14 days post-denervation, not a longitudinal observation. Consequently, to avoid confusion, we have removed the phrase "over time" from our description. We hope this clarification addresses your query.

9. In Extended Data Fig. 6, SDH staining of denervated group appears to be selective.

Response: We selected images to represent the original data as accurately as possible. To avoid confusion, we included the original staining in the Extended Data Fig. 6.

A

B

A, B: SDH staining of TA muscle in normal and denervated GAS muscle (14 days post denervation, 40X).

Minor comments:

1. Line 301: "Figure 5c" should be "Figure 4c".

Response: We appreciate the reviewer's attention to detail. The mistake has been corrected in our revised manuscript.

REVIEWERS' COMMENTS

Reviewer #2 (Remarks to the Author):

Authors updated the manuscript in a satisfactory manner.

However, I have still a few minor comments:

- 1) In Fig. 1c there is a cluster of red dots, in between MTJ and Adipocyte, that is not followed by the statement of the type of cells the cluster represents in the UMAP graph.
- 2) Line 912: While in the figure 6c is reported only 0d and 14d, in the corresponding figure legend is still reported 0d, 7d, and 14d.
- 3) In Fig. 5g LogFC there is still "denverntion".

Once edited these errors, I recommend the manuscript for publication.

Thanks,

Carlo Serra

Reviewer #3 (Remarks to the Author):

The authors responded satisfactorily to most of the comments I had. The revised manuscript more precisely illustrates the cis-regulatory network in muscle atrophy. This is a meaningful study for muscle regeneration. The existing results prove that ELK4 is a transcription factor that plays a key regulatory role in muscle atrophy, and it works through the classic TGF β pathway that has a great contribution to gene therapy for muscle atrophy. It provides effective data for the study of single-cell RNA-Seq and ATAC-Seq on muscle tissue, and it supplies a variety of data and methods for omics association analysis and mining functional genes for skeletal muscle regeneration research.

REVIEWER COMMENTS

Reviewer #2 (Remarks to the Author):

Authors updated the manuscript in a satisfactory manner.

However, I have still a few minor comments:

1) In Fig. 1c there is a cluster of red dots, in between MTJ and Adipocyte, that is not followed by the statement of the type of cells the cluster represents in the UMAP graph.

Response: Thank you for pointing this out. We have now labeled the cluster of macrophages in Fig.1c as you suggested. Please see the updated Figure 1c.

2) Line 912: While in the figure 6c is reported only 0d and 14d, in the corresponding figure legend is still reported 0d, 7d, and 14d.

Response: Thank you for highlighting this inconsistency. We apologize for the oversight and have now updated the figure legend to accurately reflect "denervated (14 days)" as shown in Figure 6c.

3) In Fig. 5g LogFC there is still "denverntion".

Response: Thank you for pointing out the typo. We regret the oversight and have now corrected "denverntion" to "Denervation" in Fig.5g.

Reviewer #3 (Remarks to the Author):

The authors responded satisfactorily to most of the comments I had. The revised manuscript more precisely illustrates the cis-regulatory network in muscle atrophy. This is a meaningful study for muscle regeneration. The existing results prove that ELK4 is a transcription factor that plays a key regulatory role in muscle atrophy, and it works through the classic TGF β pathway that has a great contribution to gene therapy for muscle atrophy. It provides effective data for the study of single-cell RNA-Seq and ATAC-Seq on muscle tissue, and it supplies a variety of data and methods for omics association analysis and mining functional genes for skeletal muscle regeneration research.

Response: Thank you for the reviewer's appreciative remarks and acknowledgment of the significance of our work. We are pleased that our study resonates and contributes to the field of skeletal muscle regeneration. We remain dedicated to furthering research in this area using the data presented in our manuscript.